# QUESTA: EXPANDING REASONING CAPACITY IN LLMS VIA QUESTION AUGMENTATION

**Jiazheng Li**[* † 1,2], **Hongzhou Lin**[* ‡ 3], **Hong Lu**[1,2], **Kaiyue Wen**[4], **Zaiwen Yang**[1],
**Jiaxuan Gao**[1], **Yi Wu**[1,2], **Jingzhao Zhang**[1,2]
[1]Tsinghua University, [2]Shanghai Qi Zhi Institute, [3]Amazon, [4]Stanford University

## ABSTRACT

Reinforcement learning (RL) has emerged as a central paradigm for training large language models (LLMs) in reasoning tasks. Yet recent studies (Yue et al., 2025; Liu et al., 2025) question RL's ability to incentivize reasoning capacity beyond the base model. This raises a key challenge: how can RL be adapted to solve harder reasoning problems more effectively? To address this challenge, we propose a simple yet effective strategy via *Question Augmentation*: introduce partial solutions during training to reduce problem difficulty and provide more informative learning signals. Our method, QuestA, when applied during RL training on math reasoning tasks, not only improves pass@1 but also pass@k—particularly on problems where standard RL struggles to make progress. This enables continual improvement over strong open-source models such as DEEPSCALER and OPENMATH NEMOTRON, further enhancing their reasoning capabilities. We achieve new state-of-the-art results on math benchmarks using 1.5B-parameter models: 72.50% (+10.73%) on AIME24, 62.29% (+12.79%) on AIME25, and 41.67% (+10.11%) on HMMT25. Code, data and model are available at `https://github.com/foreverlasting1202/QuestA`.

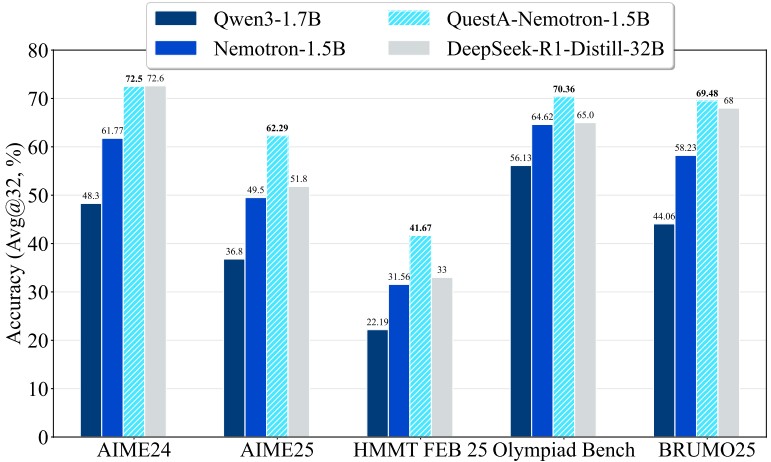

Figure 1: QUESTA is a data augmentation method that injects partial solutions to effectively scaffold RL training on hard reasoning problems. We construct 26K high-quality augmented prompts from challenging instances in OpenR1 (Open-R1 Team, 2025), and fine-tune models using 32K-context-length RL. When applied to `Nemotron-1.5B`, QUESTA delivers substantial performance gains—achieving new state-of-the-art results across all math benchmarks for 1.5B-parameter models.

## 1 INTRODUCTION

Frontier large language models (LLMs), including OpenAI-O1, O3 (Jaech et al., 2024), DeepSeek-R1 (Guo et al., 2025), Qwen3 (Yang et al., 2025), and Gemini 2.5 (Gemini Team, Google DeepMind,

---

[*]Equal Contribution
[†]Contact: Jiazheng at foreverlasting1202@outlook.com, and Jingzhao at jingzhaoz@mail.tsinghua.edu.cn
[‡]This work is independent of and outside of the work at Amazon.

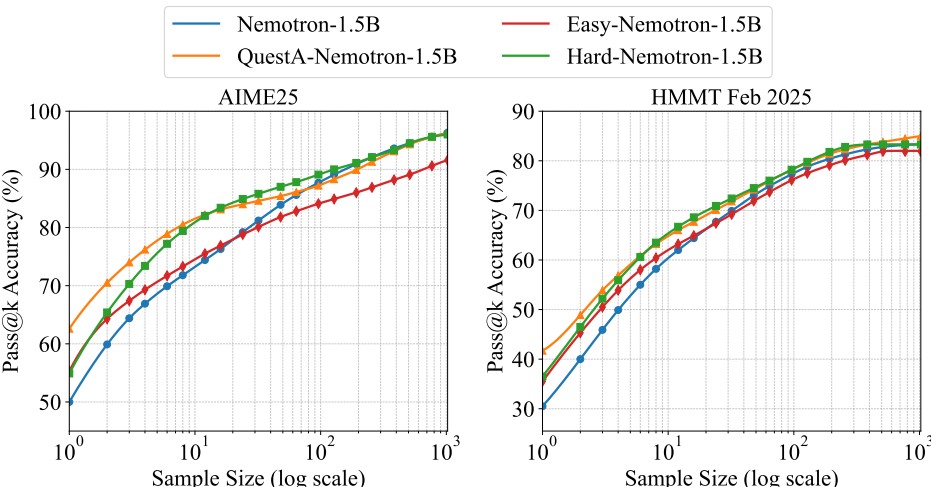

Figure 2: We compare pass@k curves of RLVR-trained models, with and without QUESTA. As a controlled experiment, we perform RL training using either easy or hard prompts. Standard RL on easy prompts (red) shows clear degradation in pass@k as $k$ increases compared to the base model (blue). Training on hard prompts (green) improves pass@k, but comes at the cost of substantially longer training. This motivates our development of QUESTA, which scaffolds hard problems to improve training efficiency and delivers consistently stronger results: the RL+QUESTA model (orange) stays above standard RL (red) across all $k$, while also preserving or improving performance at larger $k$ relative to RL trained with hard prompts.

2025), have exhibited exceptional performance on high-complexity reasoning tasks spanning mathematics, programming, and formal logic. Recent advances in the field have increasingly prioritized reinforcement learning paradigms (RL), among which *Reinforcement Learning with Verifiable Rewards* (RLVR) has emerged as a scalable and efficient approach to enhancing reasoning capabilities. Using automatically verifiable signals, RLVR enables alignment between model output and objective correctness, thus addressing a critical limitation of traditional RL for reasoning.

However, the community remains divided on a fundamental question regarding RLVR: does it expand the model's intrinsic reasoning capacity, or merely exploit pre-existing knowledge encoded in the base model? Recent research (Yue et al., 2025; Liu et al., 2025; Zhao et al., 2025) show that while state-of-the-art RL methods (e.g., GRPO, DAPO) (Guo et al., 2025; Yu et al., 2025; An et al., 2025) can enhance the pass@1 metric by reinforcing high-reward completions, they encounter significant limitations when tackling high-difficulty tasks where the base model performs poorly. This phenomenon differs from that observed in Supervised Fine-Tuning (SFT) Luo et al. (2025a). Within the SFT paradigm, enhancing the diversity of problem difficulty serves as a critical factor, as it can effectively improve the model's performance on downstream tasks. However, in the framework of RLVR, the inclusion of easy prompts tends to undermine the model's inherent reasoning capabilities.

One insightful explanation (Cui et al., 2025; Wang et al., 2025a) for the drop suggests that model overfits on correct solutions and hence causes entropy collapse, limiting its ability to explore. To validate this, we design a controlled setup that separates prompts into easy and hard groups. When applying RLVR on the Nemotron 1.5B model (Moshkov et al., 2025) with the OpenR1 dataset, we find that training on easy prompts leads to a clear decline in pass@k accuracy (Figure 2).

Given these findings, we observe that training with hard prompts is more beneficial than with easy ones. Yet, RL training on hard problems tends to be much slower, as sparse reward signals and limited sample efficiency hinder progress. The key challenge, then, is **how to structure the learning process to fully expand reasoning capabilities while mitigating the inefficiency of RL on hard tasks**. To this end, we introduce QUESTA: a parsimonious and efficient strategy that dynamically adjusts problem difficulty during RL training. The core contributions of this work are threefold:

- We notice that the evolution of model capacity in RLVR critically depends on dataset difficulty, underscoring the importance of training on *hard problems* to expand reasoning ability.

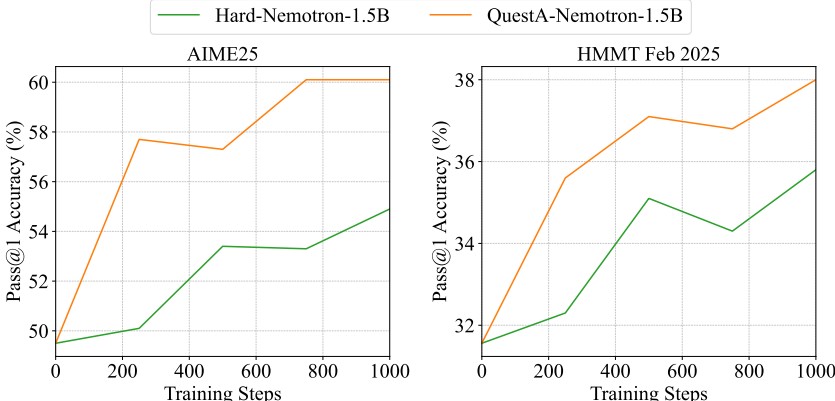

Figure 3: Comparison of RL training dynamics: Training with only hard problems (green) makes progress very slowly due to sparse rewards, while our method with partial solutions (orange) accelerates training and consistently achieves higher accuracy across training steps.

- We introduce QUESTA, an efficient procedure that controls difficulty by augmenting hard problems with partial solutions. This approach provides a smooth curriculum within RL training and makes high-difficulty tasks more tractable. Through our **fully open-sourced** training pipeline, QUESTA consistently improves pass@1 and pass@k, enabling 1.5B-parameter models to reach new state-of-the-art performance—72.5% on AIME24, 62.3% on AIME25, and 41.7% on HMMT25 (Table 1).

- Our theoretical analysis in Section 4 explains why partial-solution augmentation accelerates RL training: by decomposing problems into intermediate steps, the method yields denser reward signals and improves sample efficiency, while still driving the model to master the hardest problems.

## 2 Tradeoffs between Reasoning Capacity and Learning Efficiency

Given the ongoing debate on whether reinforcement learning enhances the reasoning capacity of language models, we design a controlled experiment to study how dataset difficulty changes model performances measured by pass@k accuracy. Specifically, we filter out easy problems and hard problems from the 220K OpenR1 dataset, base on model's success rate, each containing around 4K data. We then run RL with GRPO for one thousand steps. This setup allows us to isolate how the choice of prompt difficulty impacts the model's reasoning capacity. In Figure 2 and Figure 3, we provide pass@k comparison and the learning dynamics, we make two observations.

**RL with Easy Prompts Hurts pass@$k$ and Reasoning Capacity.** Training on easy or already-solvable problems leads to overfitting on shallow patterns, reinforcing confidence rather than expanding reasoning capacity. While pass@1 may rise, output diversity declines and performance on harder benchmarks deteriorates, with pass@$k$ dropping at larger $k$ (see Figure 2). This suggests that the model exploits familiar solution modes instead of exploring new trajectories. To truly expand capacity, RL training should focus on *hard* problems, where the policy is forced to explore and acquire novel solution strategies.

**RL with Hard Prompts Leads to Slow Learning.** Training on hard prompts directly targets the reasoning capacity of the model, but the learning process is much slower (see Figure 3) and less sample-efficient. The difficulty arises because RL rewards on these problems are sparse, providing limited gradient signals for policy improvement. We formallize the underlying reason in Section 4 and in Theorem 4.4.

In practice, not all questions in the training set $\mathcal{Q}$ are equally difficult, and one might hope that training on easier examples could generalize to harder ones. However, empirical evidence suggests that RL-based training exhibits a bi-modal pattern in success rates (An et al., 2025): by the end of training, models tend to either solve a question reliably or fail entirely (see Figure 6). This implies that once a question falls outside the model's capacity set, the RL algorithm is unlikely to recover.

Together, these results highlight a tension: *easy prompts dilute reasoning capacity, while hard prompts stall learning altogether.* This motivates the need for strategies that can retain the benefits of hard problems while mitigating the inefficiency caused by sparse rewards. To this end, we introduce partial solutions that break a complex question into smaller, more approachable pieces. Theoretical analysis (Theorem 4.6) suggests that appending part of the solutions as hint can greatly improve RL efficiency.

Empirically, we simply choose the hint to be a part of the solution of the original question $q$ and observe faster learning in Figure 3. Surprisingly, even if we don't explicitly train the model to generate the hint, the model's capacity without hint still continues to improve and lead to steady improvement in problems out of reach in standard RL training (see Table 3). We elaborate on implementation details in the next section.

## 3 QUESTA: QUESTION AUGMENTATION WITH PARTIAL SOLUTIONS

QUESTA is a modular augmentation framework designed to inject partial solution sketches into prompts during reinforcement learning (RL) training. It adresses scenarios where the base model fails to generate correct completions—conditions that typically result in sparse reward signals. Distinct from approaches that modify reward functions or optimization algorithms, QUESTA operates at the input level: it transforms original training prompts into more tractable variants, thereby exposing intermediate reasoning steps to the model.

---

**Original Prompt**

Let $\mathbb{N}$ be the set of positive integers. A function $f : \mathbb{N} \to \mathbb{N}$ satisfies the equation

$$f(f(\dots f(n)\dots)) = \frac{n^2}{f(f(n))} \quad \text{with } f(n) \text{ applications of } f,$$

for all positive integers $n$. Given this information, determine all possible values of $f(1000)$.

---

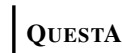 **QUESTA**

---

**Augmented Prompt**

Let $\mathbb{N}$ be the set of positive integers. The function $f : \mathbb{N} \to \mathbb{N}$ satisfies the equation

$$f(f(\dots f(n)\dots)) = \frac{n^2}{f(f(n))} \quad \text{with } f(n) \text{ applications of } f,$$

for all positive integers $n$. Given this information, determine all possible values of $f(1000)$.

**## Hint: Partial Solution**

Analysis shows that $f$ must be an involution, meaning $f(f(n)) = n$ for all $n$, and it fixes all odd positive integers, so $f(n) = n$ for odd $n$. For even positive integers, $f$ either fixes the number or swaps it with another even positive integer in a 2-cycle.

Please reason step by step, and put your final answer within \boxed{ }.

---

Figure 4: QUESTA augments each original question in the dataset by prepending the first $p\%$ of the solution sketch. In our experiments, we apply augmentation using the solution block rather than the reasoning chain-of-thought. The hint percentage $p$ is computed as the ratio of tokens used as hints to the total number of tokens in the solution sketch.

**Question Augmentation Mechanism** For a given problem $x$ with an $n$-step solution trajectory $y = (y_1, y_2, \dots, y_n)$, QUESTA constructs a set of augmented prompts $\{\tilde{x}^{(p)}\}$, where each $\tilde{x}^{(p)}$ appends the first $p$ steps of the solution as a prefix to the original question. The parameter $p$ (e.g., $p = 50\%$ or $25\%$) quantifies the proportion of the solution revealed, thereby enabling precise control over the difficulty of the augmented prompt.

In our empirical evaluations, we employed the OpenR1-Math-220K dataset (Open-R1 Team, 2025)—a supervised fine-tuning (SFT) corpus containing solution trajectories generated by DEEPSEEK-R1. Each instance in this dataset comprises a detailed chain-of-thought (CoT) section followed by a final solution block. For augmentation, we extracted the final solution (omitting speculative reasoning

within the CoT section). The solution was then truncated at a predefined percentage $p$ and prepended to the original question, yielding the augmented prompt used in RL training, as shwon in Figure 4.

**Targeting High-Difficulty Problems**     QUESTA is applied exclusively to prompts where the base model's pass rate is close to zero. Using the OpenR1-Math-220K dataset, we first employ lightweight heuristic filters to reduce the full 220K problems to 26K of the hardest candidates. These problems are then augmented with partial-solution prefixes where we conduct a second difficulty screening: sample multiple completions from the model for each augmented prompt, and only those instances with consistently low pass rates are retained. This two-stage filtering pipeline yields a final pool of no more than 10K problems, ensuring that augmentation resources are concentrated on the most challenging cases where the base model needs additional guidance and scaffolding.

**Integrating with RL Pipelines**     QUESTA exhibits orthogonality to underlying RL algorithms, enabling seamless integration into existing training pipelines (e.g., GRPO (Shao et al., 2024), DAPO (Yu et al., 2025)) without modifications. Specifically, integration requires only replacing the original rollout dataset with the augmented dataset, while retaining the original reward function and policy update mechanism. To further exploit this input-level flexibility, we extended QUESTA with an iterative curriculum RL paradigm:

1. First, augment the dataset with $p = 50\%$, apply the difficulty filtering with the augmented prompt, and conduct reinforcement learning training until the performance saturates.

2. Second, reduce the augmentation from $p = 50\%$ to $p = 25\%$, i.e. provide fewer hints. Again, we apply the difficulty filtering, and conduct reinforcement learning training until convergence.

Here, the rationale for the choice of $p$ is provided in Appendix B.6. By keeping the training signals strong at each stage, the method speeds up convergence on difficult tasks and makes QUESTA a simple, plug-and-play approach for curriculum-based RL.

## 4  THEORY: VARYING LEARNABILITY ENHANCES RL EFFICIENCY

In this section, we present a theoretical perspective on how question augmentation improves the efficiency of reinforcement learning. Our central thesis is that the primary bottleneck in RL-based reasoning lies in the difficulty of discovering successful trajectories within a finite sampling budget. Question augmentation addresses this challenge by reshaping the *learnability landscape*—making hard problems more discoverable by increasing the likelihood of encountering correct trajectories.

Motivated by experiments which quantify model capcity with pass@k accuracy, we introduce the following notions of *solution set* (Definition 4.1) and *model capacity set* (Definition 4.2) for a given question $q$ and model $\mu$. Let $\mathcal{V}$ be the vocaboluary set, and let $P_\mu(q, \tau)$ denote the probability that a language model $\mu$ generates trajectory $\tau \in \mathcal{V}^*$ when conditioned on input question $q \in \mathcal{V}^*$.

**Definition 4.1** (Solution Set). Given a question $q$ and a binary reward function $\mathrm{R} : \mathcal{V}^* \times \mathcal{V}^* \to \{0, 1\}$, the *solution set* is defined as:

$$\mathcal{S}(q) = \{\tau \in \mathcal{V}^* \mid \mathrm{R}(q, \tau) = 1\}.$$

**Definition 4.2** (Model Capacity Set). Given a probability threshold $\delta_p > 0$, a language model $\mu$, and a question $q$, define the *model capacity set* $C(q, \delta_p)$ as the smallest set of trajectories whose total probability mass is at least $1 - \delta_p$:

$$C(q, \delta_p) = \arg \min_{S \subseteq \mathcal{V}^*} \left\{ |S| \;\middle|\; \sum_{\tau \in S} P_\mu(q, \tau) \geq 1 - \delta_p \right\}.$$

The *Model Capacity Set* $C(q, \delta_p)$ intuitively captures the set of most likely output trajectories that the model $\mu$ can generate for a given input $q$, up to a small probability threshold $\delta_p$.

This formalization leads to a critical insight: if the model's capacity set fails to intersect with the solution set—meaning the model is unlikely to generate any correct completions—then the RL process cannot make progress. To articulate this more formally, we begin by stating a standard assumption satisfied by many popular RL algorithms, such as DAPO and online GRPO:

**Assumption 4.3** (Null Gradient from Zero-Reinforcement). The RL algorithm does not update the model weights if none of the sampled rollouts receives a positive reward (i.e., reward = 1).

Under this assumption, we easily the following lower bound, which states that if all training questions are unreachable within the model's capacity set, the RL process is likely to stall entirely:

**Theorem 4.4** (Lower Bound on RL Learnability under Solution Inaccessibility). *Given a probability threshold $\delta_p > 0$, if for every question $q \in \mathcal{Q}$, the model capacity set $C(q, \delta_p)$ does not intersect with the solution set $\mathcal{S}(q)$, i.e.,*

$$C(q, \delta_p) \cap \mathcal{S}(q) = \emptyset, \quad \forall q \in \mathcal{Q},$$

*then under Assumption 4.3, when training RL for $T$ steps with $B$ samples per step such that $TB = \Theta(1/\delta_p)$, there is a constant probability that the RL algorithm will not update the model.*

To overcome this limitation, our method QUESTA provides a simple yet effective solution: augment each question in $\mathcal{Q}$ with a partial solution to improve the chances of sampling informative trajectories. Formally, we assume the existence of a hint $h_q$ for every question $q \in \mathcal{Q}$ that can guide the model toward discovering a valid completion.

**Definition 4.5** (Question Augmentation). For every question $q \in \mathcal{Q}$, hint $h_q \in \mathcal{V}^*$ satisfies that for $\delta'_p = \delta_p^{1/2-\epsilon}$ for some $\epsilon > 0$:

- the hint $h_q$ can be generated with a non-neglible probability: $P_\mu(h_q|q) \geq \delta'_p$.

- there exists a solution to the hinted problem $s_q \in \mathcal{S}(q)$ such that $s_q$ can be generated with high probability after $s_q$, i.e.

$$P_\mu(s_q|(q, h_q)) = \delta'_p. \quad R(q, h \oplus s_q) = 1.$$

The hint $h_q$ can exist for every question even when the model's capacity set $C(q, \delta_p)$ does not intersect with the solution set $\mathcal{S}(q)$. For instance, if every solution can be decomposed into two steps, and the model can generate each step correctly with probability $\delta'_p = \sqrt{o(\delta_p)}$, then the possibility of generating two steps correctly at the same time is only $o(\delta_p)$.

This implies that the sampling budget needed with a hint is asymptotically almost the square root of the budget required without it ($\Theta(1/\delta_p)$), as given in Theorem 4.4. We further provide a learnability result where we assume the policy is parameterized by a softmax policy parameterization in a classical tabular RL setup.

**Theorem 4.6** (Informal Upper Bound on RL Learnability with Hint). *If we have a hint $h_q$ for every question $q \in \mathcal{Q}$ (Def. 4.5), then there exists an RL algorithm that can output a policy $\pi_\theta$ such that $\mathbb{E}_{q \sim \text{Uniform}(\mathcal{Q})}[\mathbb{P}_{\tau \sim \pi_\theta(\cdot|q)}(\tau \in \mathcal{S}(q))] \geq 0.99$ with $O(1/\delta'_p)$ sampling budget with high probability.*

Theorem 4.6 provides a theoretical guarantee that the model can reach a high training success rate when partial solution is included. Empirically, we observe the model generalizes well both in-distribution and out-of-distribution to hard questions.

## 5 EXPERIMENTS

**Dataset.** We begin with the OpenR1-Math-220K dataset and use DeepSeek-R1-Distill-1.5B as a weak selection model to filter it down to the 26K hardest items. This set serves as our base prompts. We then use Nemotron-1.5B to sample eight generations per prompt and classify problems into Easy Data (7–8 correct answers) and Hard Data (0–1 correct answers), enabling controlled experiments introduced in Section 2. The exact prompt template is provided in Appendix B.8

**Data Augmentation (QUESTA).** To improve the tractability of the problems, we apply QUESTA to prepend the prompt with partial solutions, i.e. first p% of the full solution in the SFT data provided in the OpenR1-Math-220K dataset. After augmentation, we use the initial model at RL training, either Nemotron-1.5B or DeepScaleR-1.5B, to sample 8 generations per augmented prompts and select samples with 0–4 correct predictions. Full details are provided in Appendix B.5. These high-variance cases provide stronger learning signals and make the training process more effective.

Table 1: Performance comparison (Pass@1, averaged over 32 samples) across maths benchmarks. The best results among the 1.5B models are highlighted in bold. Larger models are shown in gray as reference points. Reported results for DeepSeek-R1-Distill and Qwen3 are taken from their official documentation (Guo et al., 2025; Yang et al., 2025), while the rest are self-evaluated. Our QUESTA-Nemotron-1.5B achieves state-of-the-art performance among 1.5B models and, notably, matches or even exceeds the performance of DeepSeek-R1-Distill-32B across several benchmarks, despite being over 20× smaller in parameter count. This demonstrates the effectiveness of QUESTA in enhancing small model capabilities through targeted training.

| Model | AIME24 | AIME25 | HMMT FEB 25 | Olympiad Bench | BRUMO25 | Avg |
|---|---|---|---|---|---|---|
| DeepSeek-R1-Distill-1.5B | 28.7 | 22.3 | 12.0 | 52.4 | 31.8 | 29.44 |
| Qwen3-1.7B | 48.3 | 36.8 | 22.19 | 56.13 | 44.06 | 41.50 |
| DeepSeek-R1-Distill-32B | 72.6 | 51.8 | 33 | 65.0 | 68 | 58.08 |
| Qwen3-8B | 76.0 | 67.3 | 44.79 | 68.56 | 68.33 | 64.99 |
| Nemotron-1.5B | 61.77 | 49.50 | 31.56 | 64.62 | 58.23 | 53.14 |
| QUESTA-Nemotron-1.5B | **72.50** | **62.29** | **41.67** | **70.36** | **69.48** | **63.26** |

**Training Setup.** We use AReaL (Fu et al., 2025) as our RL training framework, applying the GRPO algorithm (Guo et al., 2025) without the Kullback–Leibler (KL) divergence loss. Following DAPO (Yu et al., 2025), we also dynamically filter out prompts that are either all correct or all incorrect during rollouts. During training, we sample $n = 16$ responses per prompt with a maximum prompt length of 8192 tokens and a maximum generation length of 24000 tokens, using a sampling temperature of 1.0 and clipping hyperparameters with $\varepsilon_{\text{low}} = \varepsilon_{\text{high}} = 0.2$. The batch size is 128 with a mini-batch size of 1, equivalent to 128 gradient updates per rollout step. Optimization is performed with AdamW (Kingma & Ba, 2017; Loshchilov & Hutter, 2019) using a constant learning rate of $2 \times 10^{-5}$. Experiments are conducted on eight NVIDIA H800 (80GB) nodes. Full details of our training method are provided in Appendix B.1.

**Evaluation Setup.** For each problem in the evaluation benchmarks, we generate 32 samples and report pass@1 results. Generation uses a sampling temperature of 0.7 and a top-$p$ value of 0.95, with $k = 32$ responses per question unless otherwise specified. *It is important to note that while partial solutions were incorporated during training, no partial solutions are provided at evaluation time.*

## 5.1 EXPERIMENTAL RESULTS

**Key Results.** Table 1 reports results on challenging math benchmarks. QUESTA yields substantial gains for Nemotron-1.5B, achieving an average improvement of $10\%$ over its baseline and a particularly strong **+13%** on AIME25. These improvements are consistent across all benchmarks, highlighting the effectiveness of our approach in enhancing problem-solving robustness.

Compared to other models, QUESTA-Nemotron-1.5B consistently outperforms peers of similar scale, such as DeepSeek-R1-Distill-1.5B and Qwen3-1.7B, and even surpasses larger models like DeepSeek-R1-Distill-32B across all benchmarks. On AIME25 in particular, it exceeds DeepSeek-R1-Distill-32B by a substantial margin of $+11\%$. Against the stronger Qwen3-8B, QUESTA-Nemotron-1.5B remains competitive despite operating at a fraction of the parameter scale.

**Training Dynamics.** Figure 5 summarizes the training dynamics of QUESTA-Nemotron-1.5B. A positive correlation is observed between average response length and model accuracy, reflecting common trends in RL training. Notably, with QUESTA, the entropy during RL training remains stable and does not exhibit significant collapse.

**Pass@k Analysis.** Our evaluation follows the standard pass@$k$ methodology, consistent with DeepSeek-R1 (Guo et al., 2025), with further details provided in Appendix B.2. In contrast to recent findings that RL-based training can reduce pass@$k$ at larger $k$ values (Yue et al., 2025; Liu et al., 2025), our results show that QUESTA preserves—and in many cases modestly improves—performance across a broad range of $k$. As shown in Figure 2, incorporating partial-solution hints within a two-stage curriculum yields consistent gains across models, without the degradation in pass@$k$ often observed under standard RL training. These results indicate that QUESTA enhances both the quality and diversity of candidate solutions, rather than overfitting to a single best trajectory.

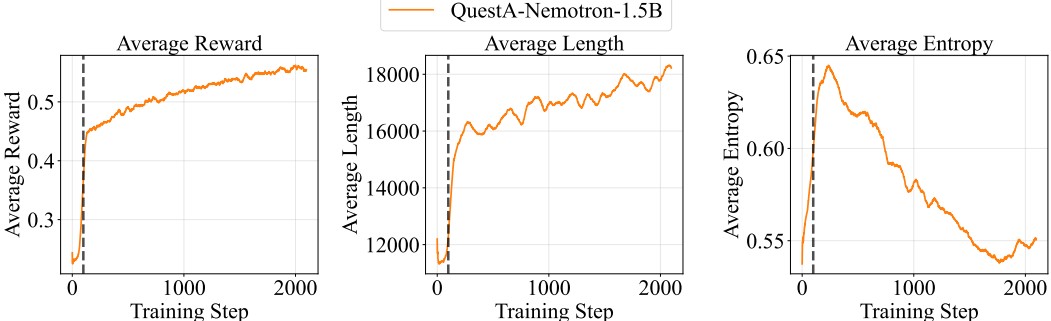

Figure 5: Training dynamics of QUESTA-Nemotron-1.5B. The first and second charts show the progression of average response length and average reward across rollout samples during the RL process, both of which steadily increase over time. The third chart presents the average entropy. Interestingly, the entropy increases over time, suggesting that QUESTA does not suffer from entropy collapse and instead encourages diverse and exploratory behavior.

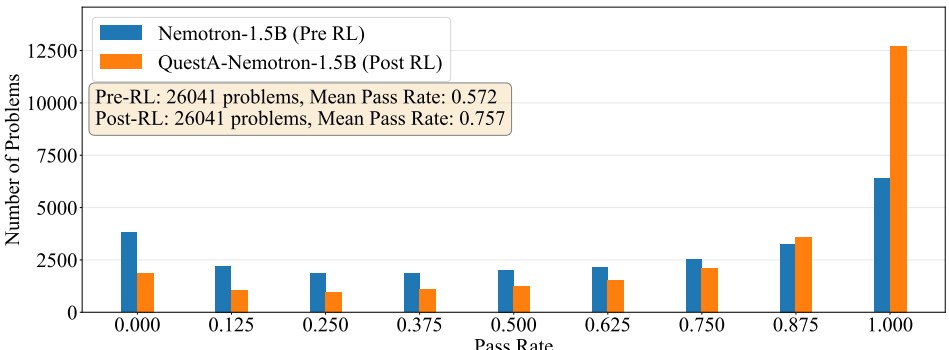

Figure 6: Pass Rate Distribution on Training Prompts. We compare the success rate on the 26K training set before and after RL, using the average pass rate over 8 samples per question. Although partial solutions are included during QUESTA training, **no hints** are provided during this evaluation. This setup isolates the true impact of QUESTA by assessing its ability to improve performance on problems without hints. QUESTA significantly reduces the number of unsolved or partially solved problems in the training set, especially for hard ones where initial model solves only 0/8 or 1/8 times.

**Generalization at Test Time when Hints are Removed.** A natural question arises from our approach: since we add partial solutions during RL training, does this improvement persist when hints are removed at evaluation time? To answer this, Figure 6 compares the pre- and post-RL models on the 26K training prompt set, evaluated without any hints. The distribution clearly shifts away from the 0/8–1/8 bins toward higher pass rates, indicating that the model solves a larger fraction of problems even without access to partial solutions. On the evaluation AIME benchmarks, Table 2 further demonstrates that QUESTA expands coverage at `Pass@32`: for `Nemotron-1.5B`, the number of unsolved problems drops from 5 to 2 on AIME24 (newly solved indices 2, 13, 29) and from 6 to 3 on AIME25 (newly solved indices 9, 13, 27). Taken together, these results show that our method generalizes well beyond the training setting and helps solve hard problems that are otherwise inaccessible without partial-solution guidance.

## 5.2 FURTHER ABLATIONS

**Ablation with Difficulty Curriculum.** We first motivate the choice of a two-stage curriculum: RL on *Partial-50* followed by RL on *Partial-25*. From a modeling standpoint, the most appropriate inference distribution for the model should be the original (no-hint) distribution. Hence, during training we should gradually reduce reliance on hints to align the learned policy with the evaluation distribution. This motivates decreasing the partial ratio over time so that the model transitions from scaffolded reasoning to autonomous reasoning.

Table 2: Indices of **unsolved** problems at `Pass@32` on AIME24 and AIME25 (with indices ranging from 0-29). Our method, QUESTA, consistently improve the model capacity on hard cases where the initial model is unable to solve, improving overall coverage at `Pass@32`.

| Models | AIME24 Unsodlved Indices | AIME25 Unsolved Indices |
|---|---|---|
| Nemotron-1.5B | 2, 3, 13, 21, 29 | 9, 12, 13, 14, 27, 29 |
| QUESTA-Nemotron-1.5B | 3, 21 | 12, 14, 29 |

Table 3: Ablation Study on the Impact of Curriculum Design. This table demonstrates the importance of curriculum learning in improving model performance. The model QUESTA-Nemotron-1.5B-50 was trained entirely with Partial-50 data for 2000 steps, while QUESTA-Nemotron-1.5B followed a curriculum learning approach, starting with 100 steps of Partial-50 data followed by 1900 steps of Partial-25 data. As seen in the table, the curriculum learning approach (QUESTA-Nemotron-1.5B) outperforms training with only Partial-50 data (QUESTA-Nemotron-1.5B-50). Extension with Partial-50→Partial-25→Partial-0 did not yield significant improvements, and thus, are not included in the table.

| Model | AIME24 | AIME25 | HMMT FEB 25 | Olympiad Bench | BRUMO25 | Avg |
|---|---|---|---|---|---|---|
| Nemotron-1.5B | 61.77 | 49.50 | 31.56 | 64.62 | 58.23 | 53.14 |
| QUESTA-Nemotron-1.5B-50 | 67.18 | 59.38 | 39.17 | 69.41 | 66.15 | 60.26 |
| QUESTA-Nemotron-1.5B | **72.50** | **62.29** | **41.67** | **70.36** | **69.48** | **63.26** |

Empirically, Table 3 shows that, under the same 2000-step budget, the curriculum *Partial-50→Partial-25* learns substantially better than training on *Partial-50* alone. We cap the *Partial-50* stage at 100 steps, after which we switch to *Partial-25*. As shown in Figure 11, entropy for QUESTA-Nemotron-1.5B-50 begins to decline beyond 100 steps, so transitioning at this point prevents overconfidence and sustains training stability. We have also tried extending the curriculum from *Partial-25* to *Partial-0* in our experiments, but observed no gains and no increase in response length (see Figure 12).

Table 4: Performance comparison (Pass@1, averaged over 32 samples) between Nemotron-1.5B and QUESTA-Nemotron-1.5B (By OpenMathReasoning). The two models achieve comparable results, with the version trained on OpenR1 performing slightly better overall.

| Model | AIME24 | AIME25 | HMMT FEB 25 | Olympiad Bench | BRUMO25 | Avg |
|---|---|---|---|---|---|---|
| Nemotron-1.5B | 61.77 | 49.50 | 31.56 | 64.62 | 58.23 | 53.14 |
| QUESTA-50 (with OpenMathReasoning) | 66.46 | 58.54 | 36.35 | 66.06 | 63.13 | 58.11 |
| QUESTA-50 (with OpenR1) | 67.18 | 59.38 | 39.17 | 69.41 | 66.15 | 60.26 |

**Ablation with Different Dataset.**   We also evaluated QUESTA on OpenMathReasoning Moshkov et al. (2025), selecting the 60K questions with `pass_rate_72b_tir` of 0 or 1/32. Due to time constraints, we trained only the first stage of QUESTA with 50% partial solutions. Table 4 shows that QUESTA-Nemotron-1.5B-50 achieves similar performance as using the OpenR1 dataset. This indicates that our approach generalizes across datasets.

**Other Ablations.**   We also conduct an extensive set of comparative experiments and ablation studies, with detailed results provided in Appendix D. These include an ablation of QUESTA without hints (Appendix D.1), experiments with different model backbones (Appendix D.2), and the full set of pass rates and training curves for additional models (Appendix D.3).

## 6  CONCLUSIONS

In this work we introduced QUESTA, a lightweight data-centric framework that augments hard prompts with partial-solution hints during RL training. Without altering model architecture or reward design, QUESTA sets new state-of-the-art results for 1.5 B-scale models on AIME24, AIME25 and HMMT25. Further, we theoretically demonstrate how question augmentation can improve sample

efficiency. Our analysis shows that the method can potentially be generalized to other domains such as competitive coding, software engineering or other agentic tasks. Desigining proper question augmentation pipelines for theses new tasks can be an important and interesting future direction.

## 7 ETHICS STATEMENT

We use only public, non-PII datasets—OpenR1-Math-220K (Apache 2.0) and OpenMathReasoning (CC BY 4.0)—in full compliance with their licenses (including attribution and modification notices); no new human-subjects data were collected, no re-identification was attempted, and no IRB review was required. Our augmentation pipeline generates math problems and solutions while avoiding harmful or copyrighted non-math content; outputs may inherit source biases, so we report settings transparently, discourage high-stakes deployment or misuse without safeguards and human oversight, and will release artifacts that respect the original licenses.

## 8 REPRODUCIBILITY STATEMENT

To ensure reproducibility, we provide the code, dataset and model in the the supplementary materials and anonymous github `https://anonymous.4open.science/r/questa932/README.md`. In the README.md file included with the code, we present a step-by-step guide for reproducing our results.

## 9 ACKNOWLEDGEMENTS AND DISCLOSURE OF FUNDING

J.Z. acknowledges support by the National Key R&D Program of China 2024YFA1015800 and Xiongan AI Institute.

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

# A    THE USE OF LARGE LANGUAGE MODELS (LLMs)

The Large Language Models (LLMs) were exclusively utilized to polish the writing and detect potential typos, with no involvement in other aspects.

# B    IMPLEMENTATION DETAILS

## B.1    RLVR ALGORITHMS

We have employed the GRPO algorithm enhanced with a subset of DAPO techniques. Primarily, we have integrated DAPO's Dynamic Sampling Trick and eliminated the KL divergence term, resulting in an optimization objective that is:

$$
\begin{aligned}
\mathcal{J}(\theta) = \quad & \mathbb{E}_{q\sim\mathcal{D},\{o_i\}_{i=1}^{G}\sim\pi_{\theta_{\text{old}}}(\cdot|q)} \\
& \left[ \frac{1}{G}\sum_{i=1}^{G}\frac{1}{|o_i|}\sum_{t=1}^{|o_i|} \min\left( r_{i,t}(\theta)\hat{A}_{i,t},\ \text{clip}\left( r_{i,t}(\theta), 1-\varepsilon, 1+\varepsilon \right)\hat{A}_{i,t} \right) \right]
\end{aligned} \tag{1}
$$

$$
\text{s.t.} \quad 0 < \#\left\{ o_i \mid [o_i \text{ is correct}] \right\} < G,
$$

where

$$
r_{i,t}(\theta) = \frac{\pi_\theta(o_{i,t} \mid q, o_{i,<t})}{\pi_{\theta_{\text{old}}}(o_{i,t} \mid q, o_{i,<t})}, \quad \hat{A}_{i,t} = \frac{R_i - \text{mean}(\{R_i\}_{i=1}^{G})}{\text{std}(\{R_i\}_{i=1}^{G})}. \tag{2}
$$

Our reward function $R$ mirrors that of DeepScaleR (Luo et al., 2025b), employing an Outcome Reward Model. It returns 1 if and only if both the answer and format are correct; otherwise, it returns 0. In summary, our reward function yields:

$$
R = \begin{cases} 1, & \text{if the answer (e.g. passes basic LaTeX/Sympy checks)} \\ & \quad \text{and format (e.g. exists \texttt{<think>} and \texttt{</think>}) are both correct,} \\ 0, & \text{otherwise.} \end{cases} \tag{3}
$$

## B.2    LOW-VARIANCE PASS@k ESTIMATION

Pass@$k$ is a measure of a model's problem - solving ability, indicating the probability that the model can generate at least one correct solution in $k$ attempts. Specifically, for each problem $x_i$ in the evaluation dataset $\mathcal{D}$, we generate $n$ samples (where $n \geq k$) and count the correct ones as $c_i$. The direct calculation formula is:

$$
\text{pass@}k := \mathbb{E}_{x_i\sim\mathcal{D}}\left[ 1 - (1 - \frac{c_i}{n})^k \right] \tag{4}
$$

However, this formula has excessive variance and insufficient accuracy. To solve this problem, we adopt the unbiased estimation method proposed by Chen et al. (Chen et al., 2021), using the unbiased estimator of pass@$k$ over the dataset:

$$
\text{pass@}k := \mathbb{E}_{x_i\sim\mathcal{D}}\left[ 1 - \frac{\binom{n-c_i}{k}}{\binom{n}{k}} \right] \tag{5}
$$

In our experiments, to ensure sufficient accuracy, we set $n$ such that $2k \leq n$, which helps further reduce the variance of the estimate.

## B.3    MORE RELATED WORKS

Recent studies show that RL algorithms, such as PPO (Schulman et al., 2017) and GRPO (Guo et al., 2025), can greatly enhance model reasoning capabilities. Building on this, several works have refined this paradigm from different perspectives. One method can be adjusting the reward function. Some studies (Zhu et al., 2025; Shao et al., 2025) directly modify the reward function to improve training efficiency. Other methods introduced intermediate process rewards (Wang et al., 2024; Malik et al., 2025), while Wen et al. (Wen et al., 2025) set up a separate correctness judgment for CoT to obtain rewards.

Another novel perspective aims to improve sample efficiency by measuring certainty. For example, TreeRL (Hou et al., 2025) and VinePPO (Kazemnejad et al., 2025) enhanced sample effects by introducing entropy or confidence. MRT (Qu et al., 2025), on the other hand, reused partial trajectories during testing to boost sample efficiency. R3 (Xi et al., 2024) improves RL sample efficiency by decomposing human solution steps and providing preceding steps to guide the model in completing subsequent ones. Further, some research adopted a multi-stage training or reasoning mode, exploring from different angles such as training length (Luo et al., 2025b), question difficulty (Parashar et al., 2025), and fixed-length summaries during reasoning (Yan et al., 2025).

In addition to designing better algorithms, another line of research (Shao et al., 2024; Yue et al., 2025; Zhao et al., 2025) has investigated how reinforcement learning affects the frontier of model capabilities, observing a decay in pass@k when k becomes large. In response to this phenomenon, some works (Yu et al., 2025; Liu et al., 2025; An et al., 2025) maintained entropy stability by adjusting training entropy through methods such as increasing the clipping upper bound, enlarging the temperature coefficient, extending the training length, and periodically updating the KL reference model. StepHint (Zhang et al., 2025) also preserved entropy stability by leveraging intermediate thinking content of iterative length as a prompting signal.

In contrast to the aforementioned research, our work adopts an orthogonal approach by using part of the ground-truth solution as a hint, without requiring any modifications to the existing reinforcement learning infrastructure. We provide both theoretical justification and empirical evidence that this strategy maintains pass@k without compromising the exploratory capacity of the underlying reinforcement learning algorithm.

### B.4 BENCHMARKS

We evaluate the models' breadth across various tasks in multiple domains, including mathematics, coding, reasoning, and logical inference. For mathematics, we follow DeepScaleR (Luo et al., 2025b) and Nemotron (Moshkov et al., 2025), and conduct assessments on more challenging mathematical datasets such as AIME2024 (MAA, 2024), AIME2025 (MAA, 2025), Olympiad Bench (He et al., 2024), HMMT FEB 25 (hmm, 2025), and BRUMO25 (bru, 2025). Specifically, HMMT25 Feb and BRUMO25 are both sourced from MathArena (Balunović et al., 2025). In the realm of coding, we utilize commonly employed datasets, including Code Contests (Li et al., 2022), Codeforces[1], and LCB V5 202410-202502 (Jain et al., 2024). For logical reasoning tasks, we assess our models' capabilities using GPQA Diamond (Rein et al., 2023) [2]and Zebraliogic (Lin et al., 2025). The benchmarks related to coding and logical reasoning are all referenced from AReaL (Fu et al., 2025).

### B.5 TRAINING DATASET

The dataset employed in our study is OpenR1-Math-220K (Open-R1 Team, 2025). Prior to commencing the training of the Partial Solution, we conducted a preliminary screening of the dataset. Specifically, we utilized the DeepSeek-R1-Distill-1.5B (Guo et al., 2025) model to perform eight inference operations on each of the 220k data entries in the OpenR1 dataset. Subsequently, we compared the annotated answers in the OpenR1 dataset with the results generated from each inference to tally the number of correct instances for each data entry. Ultimately, we selected the data entries with 0 or 1 correct instance as the training samples for our study. The final dataset size is 26K.

For controlled comparisons, we further split this 26K subset by re-sampling **Nemotron-1.5B** eight times per item and counting correct completions. We define *Easy Data* as questions with correct counts in $[7, 8]$ and train a model on this split, denoted `Easy-Nemotron-1.5B`. Similarly, we define *Hard Data* as questions with correct counts in $[0, 1]$ and train `Hard-Nemotron-1.5B` on this split.

Additionally, for the augmented data, we perform eight inference passes using the model currently under training. We then select samples for which the number of correct predictions falls within the range of $[0, 4]$. This criterion is motivated by the finding that samples exhibiting higher variance are more beneficial for training (Gao et al., 2025; Wang et al., 2025b). The range $[0, 4]$ is chosen because it includes the point of maximum sample variance, which is achieved with four correct predictions out of eight trials. For convenience, we refer to augmented data with partial ratio $p$ as *Partial-p* data.

---

[1]https://codeforces.com/

[2]In the GPQA Diamond dataset, multiple-choice questions are presented in the form of options rather than directly providing the answer, requiring the model to output only A, B, C, or D.

## B.6 THE RATIONALE FOR THE CHOICE OF $p$

Table 5: Number of problems vs pass rate under different hint levels on OpenMath-Nemotron-1.5B before training. We evaluated OpenMath-Nemotron-1.5B on the OpenR1 dataset after the first round of filtering, with each problem assessed 8 times. The table illustrates the distribution of correct answers (n) where $n \in \{0, 1, \ldots, 8\}$.

| Hint Levels | 0 / 8 | 1 / 8 | 2 / 8 | 3 / 8 | 4 / 8 | 5 / 8 | 6 / 8 | 7 / 8 | 8 / 8 |
|---|---|---|---|---|---|---|---|---|---|
| *Partial-50* | 143 | 224 | 304 | 472 | 710 | 1013 | 1779 | 3655 | 17741 |
| *Partial-25* | 3155 | 1997 | 1814 | 1785 | 1902 | 2175 | 2614 | 3440 | 7159 |
| *Partial-10* | 3589 | 2090 | 1865 | 1842 | 1905 | 2176 | 2653 | 3415 | 6506 |
| *Partial-0* | 3812 | 2218 | 1854 | 1842 | 2007 | 2136 | 2517 | 3264 | 6391 |

In this study, we evaluated the performance of OpenMath-Nemotron-1.5B on the OpenR1 dataset under various hint levels. The evaluation was performed after the first round of filtering, and each problem was assessed 8 times to capture the predictive distribution. The resulting table (Table 5) shows the distribution of correct answers across different hint levels, where the values represent the number of times the model answered correctly ($n \in \{0, 1, \ldots, 8\}$).

The selection of the hint parameter $p$ was primarily based on these evaluation results. As shown in the table, the performance with a Partial-50 hint significantly reduces task difficulty, as evidenced by the high pass rates across most levels. In contrast, Partial-25 (25% hint) exhibits a performance pattern similar to that of the no-hint scenario (Partial-0), with only marginal differences in task difficulty.

This minimal difference in difficulty between Partial-0 and Partial-25 suggests that training with Partial-25 does not provide substantial gains compared to Partial-0. Consequently, we adopted a stepwise design in which the hint level is first set to $p = 50\%$, followed by $p = 25\%$, to evaluate the model's performance under varying conditions.

## B.7 EVALUATION SETUP

We configured the models to have a maximum generation length of 32,768 tokens. In line with DeepSeek-R1 (Guo et al., 2025), we utilized pass@$k$ evaluation (Chen et al., 2021), with the formula detailed in B.2. We reported pass@1 using a non-zero temperature. Specifically, we used a sampling temperature of $0.7$ and a top-$p$ value of $0.95$ to generate $k$ responses per question, typically set at 32, with deviations explicitly noted. *Particular attention should be paid to the fact that, although we incorporated partial Solution during training, it was not included in the evaluation phase.*

## B.8 DETAIL ON PROMPT TEMPLATE

> **DeepScaleR Coding's Inference:**
> < │ User │ >{input}< │ Assistant │ ><think>

> **DeepScaleR Others' Inference:**
> < │ User │ >{input}
> Please reason step by step, and put your final answer within \boxed{}.< │ Assistant │ ><think>

> **Nemotrion Coding's Inference:**
> <|im_start|>user
> {input}
> <|im_end|> <|im_start|>assistant
> <think>

> **Nemotrion Others' Inference:**
> <|im_start|>system
> Please reason step by step, and put your final answer within \boxed{}.<|im_end|>
> <|im_start|>user

> {input}<lim_endl>
> <lim_startl>assistant

---

**Training prompt with partial solutions (math RL):**
{Problem}
## Hint: {Partial Solution}
Please reason step by step, and put your final answer within \boxed{}.

---

## C   THEORY

### C.1   PROOFS

**Theorem 4.4** (Lower Bound on RL Learnability under Solution Inaccessibility). *Given a probability threshold $\delta_p > 0$, if for every question $q \in \mathcal{Q}$, the model capacity set $C(q, \delta_p)$ does not intersect with the solution set $\mathcal{S}(q)$, i.e.,*

$$C(q, \delta_p) \cap \mathcal{S}(q) = \emptyset, \quad \forall q \in \mathcal{Q},$$

*then under Assumption 4.3, when training RL for $T$ steps with $B$ samples per step such that $TB = \Theta(1/\delta_p)$, there is a constant probability that the RL algorithm will not update the model.*

*Proof.* Let $p_{\text{sol}} = \sum_{\tau^* \in \mathcal{S}(q)} P_\mu(\tau^*|q)$ denote the cumulative generation probability of any solution trajectory. By $C(q, \delta_p) \cap \mathcal{S}(q) = \emptyset$ and Def 4.2:

$$p_{\text{sol}} = \sum_{\tau^* \in \mathcal{S}(q)} P_\mu(\tau^*|q) < \delta_p$$

For $N = TB$ independent samples across $T$ steps with batch size $B$, the probability of complete failure (no solution sampled) is:

$$\mathbb{P}(\text{failure}) = (1 - p_{\text{sol}})^N > (1 - \delta_p)^N$$

Given $TB = \Theta(1/\delta_p)$, we have:

$$(1 - \delta_p)^N > (1 - \delta_p)^{\Theta(1/\delta_p)} = \Theta(1).$$

The last inequality follows from the fact that $(1 - x)^{1/x} > \exp(-1/(1 - x))$ for $x \in (0, 1)$. By Assumption 4.3, if no solution is found, the model weights remain unchanged. $\square$

**Lemma C.1** (Upper Bound on Sampling Budget for Solution Given Hint). *Given a question $q \in \mathcal{Q}$, if there exists a hint $h_q$ for the question $q$ (Def. 4.5), then if we perform $TB = \Theta(1/\delta_p') = \Theta(\delta_p^\epsilon/\sqrt{\delta_p})$ i.i.d sampling over the initial model conditioned on $(q, h_q)$, we can find a valid solution with a constant probability.*

*Proof.* By Definition 4.5, we know:

1. $P_\mu(h_q|q) \geq \delta_p'$

2. $\exists s_q \in \mathcal{S}(q) : P_\mu(s_q|(q, h_q)) \geq \delta_p'$

With $N = TB \geq 10/\delta_p'$ independent samples conditioned on $(q, h_q)$, the probability of not finding the solution $s_q$ is:

$$\mathbb{P}(\text{no solution}) = (1 - P_\mu(s_q|(q, h_q)))^N \leq (1 - \delta_p')^{10/\delta_p'} \leq \exp(-10) < 0.01.$$

Therefore, $\mathbb{P}(\text{finding solution}) > 0.99$. $\square$

**Theorem 4.6** (Informal Upper Bound on RL Learnability with Hint). *If we have a hint $h_q$ for every question $q \in \mathcal{Q}$ (Def. 4.5), then there exists an RL algorithm that can output a policy $\pi_\theta$ such that $\mathbb{E}_{q \sim \text{Uniform}(\mathcal{Q})}[\mathbb{P}_{\tau \sim \pi_\theta(\cdot|q)}(\tau \in \mathcal{S}(q))] \geq 0.99$ with $O(1/\delta_p')$ sampling budget with high probability.*

This theorem is a direct corollary of the Theorem 5 regarding the bandit setup in (Mei et al., 2022). Because the setup here is relatively simple, we also present a detailed proof for this special case here. We first formalize our setup as follows:

**Assumption C.2** (Tabular RL with Hint). We consider the tabular RL setting with softmax policy parameterization. There exists a finite set of possible questions $\mathcal{Q}$ and a finite set of possible solutions $\mathcal{S}$. For each question $q \in \mathcal{Q}$, there exists a hint $h_q$, which is a subset of solutions $h_q \subseteq \mathcal{S}$.

The policy is parameterized by a $|\mathcal{S}| \times |\mathcal{Q}|$ matrix $\theta$ in the following way:

$$\mu_\theta(s|q) = \frac{\exp(\theta_{s,q})}{\sum_{s' \in \mathcal{S}} \exp(\theta_{s',q})}$$

Here the setup is different than the autoregressive setting in our experiments and simplify the model to a tabular setup for the simplicity of analysis. We now restate the assumption on the existence of hint in this setup.

**Assumption C.3** (Hint Existence, Formal Version of Definition 4.5). For each question $q \in \mathcal{Q}$, there exists a hint $h_q \subseteq \mathcal{S}$ such that $\sum_{s \in h_q} P_\mu(s|q) \geq \delta'_p$. Further, there exists a solution $s_q \in \mathcal{S}$ such that $P_\mu(s_q|q) \geq \delta'_p \sum_{s \in h_q} P_\mu(s|q)$.

**RL Algorithm:** We will first sample $\Theta(1/\delta'_p)$ action based on the policy $\mu_\theta$ conditioned on the question $q$ and the hint $h_q$. Then we will do a one-step policy gradient update on our policy. Noted that here we can reach high reward within one step because the reward function is deterministic.

**Theorem C.4** (Formal Version of Theorem 4.6). *Under Assumption C.2 and Assumption C.3, running* 1 *steps of policy gradient update with sampling budget* $\Theta(1/\delta'_p)$, *the learned policy achieves:*

$$\mathbb{E}_{q \sim \text{Uniform}(\mathcal{Q})}[\mathbb{P}_{\tau \sim \mu_\theta(\cdot|q)}(\tau \in \mathcal{S}(q))] \geq 0.99$$

*with probability* 0.99.

*Proof.* First, by Assumption C.3, for any question $q$, we have:

$$\sum_{s \in h_q} P_\mu(s|q) \geq \delta_p \quad \text{and} \quad \exists s_q : P_\mu(s_q|q) \geq \delta'_p \sum_{s \in h_q} P_\mu(s|q)$$

With sampling budget $N = \Theta(|\mathcal{Q}|/\delta'_p)$, by Lemma C.1 and the union bound, we will find a solution $s_q$ for every question $q$ with probability at least 0.99. Suppose the found set of solutions for question $q$ is $S_q$ and all sampled solutions are $s^{(1)}, \ldots, s^{(N)}$. Then because

$$\nabla_\theta \log \mu_\theta(s|q) = e_s - \sum_{s' \in \mathcal{S}} \mu_\theta(s'|q)e_{s'}$$

We have the policy gradient being

$$\begin{aligned}
\text{PG}_{:,q} &= \frac{1}{N} \sum_{i=1}^{N} \mathbf{1}[s^{(i)} \in S_q] \nabla_\theta \log \mu_\theta(s^{(i)}|q) \\
&= \frac{1}{N} \sum_{i=1}^{N} \mathbf{1}[s^{(i)} \in S_q](e_{s^{(i)}} - \sum_{s' \in \mathcal{S}} \mu_\theta(s'|q)e_{s'}) \\
&= \frac{1}{N} \sum_{i=1}^{N} \mathbf{1}[s^{(i)} \in S_q]e_{s^{(i)}} - \left(\frac{1}{N} \sum_{i=1}^{N} \mathbf{1}[s^{(i)} \in S_q]\right)\left(\sum_{s' \in \mathcal{S}} \mu_\theta(s'|q)e_{s'}\right).
\end{aligned}$$

We can make two simple observations:

1. For every $s \notin S_q$, $\text{PG}_{s,q} < 0$.

2. There exists a $s^* \in S_q$ such that $\text{PG}_{s^*,q} > 0$.

Therefore, consider the updated parameters

$$\theta'_{s,q} = \theta_{s,q} + \eta \text{PG}_{s,q}$$

If $\eta$ is large enough, we know that $\sum_{s \in S_q} P_{\mu_{\theta'}}(s|q) \geq 0.99$. This completes the proof. $\square$

# D   ADDITIONAL EXPERIMENTAL RESULTS

## D.1   ABLATION STUDY WITHOUT HINT

Table 6: Ablation without hint on `Nemotron-1.5B`: Pass@1 (avg@32) on challenging maths benchmarks. "QUESTA-Nemotron-1.5B w/o hint" trains RL on the same data but removes hints from the prompt, while "w/ hint" uses partial-solution hints during training. With hints, the model improves all benchmarks and achieves a +2.82 average gain over *w/o hint* (63.26 vs. 60.44), on top of the improvements over the base model. The one using hint requires nearly half the number of steps compared to the one not using hint to achieve the same performance.

| Model | AIME24 | AIME25 | HMMT FEB 25 | Olympiad Bench | BRUMO25 | Avg |
|---|---|---|---|---|---|---|
| Nemotron-1.5B | 61.77 | 49.50 | 31.56 | 64.62 | 58.23 | 53.14 |
| QUESTA-Nemotron-1.5B w/o hint (2K step) | 69.48 | 59.79 | 38.85 | 68.05 | 66.04 | 60.44 |
| QUESTA-Nemotron-1.5B w/ hint (1.1K step) | 69.27 | 60.00 | 37.92 | 69.72 | 68.33 | 61.05 |
| QUESTA-Nemotron-1.5B w/ hint (2K step) | **72.50** | **62.29** | **41.67** | **70.36** | **69.48** | **63.26** |

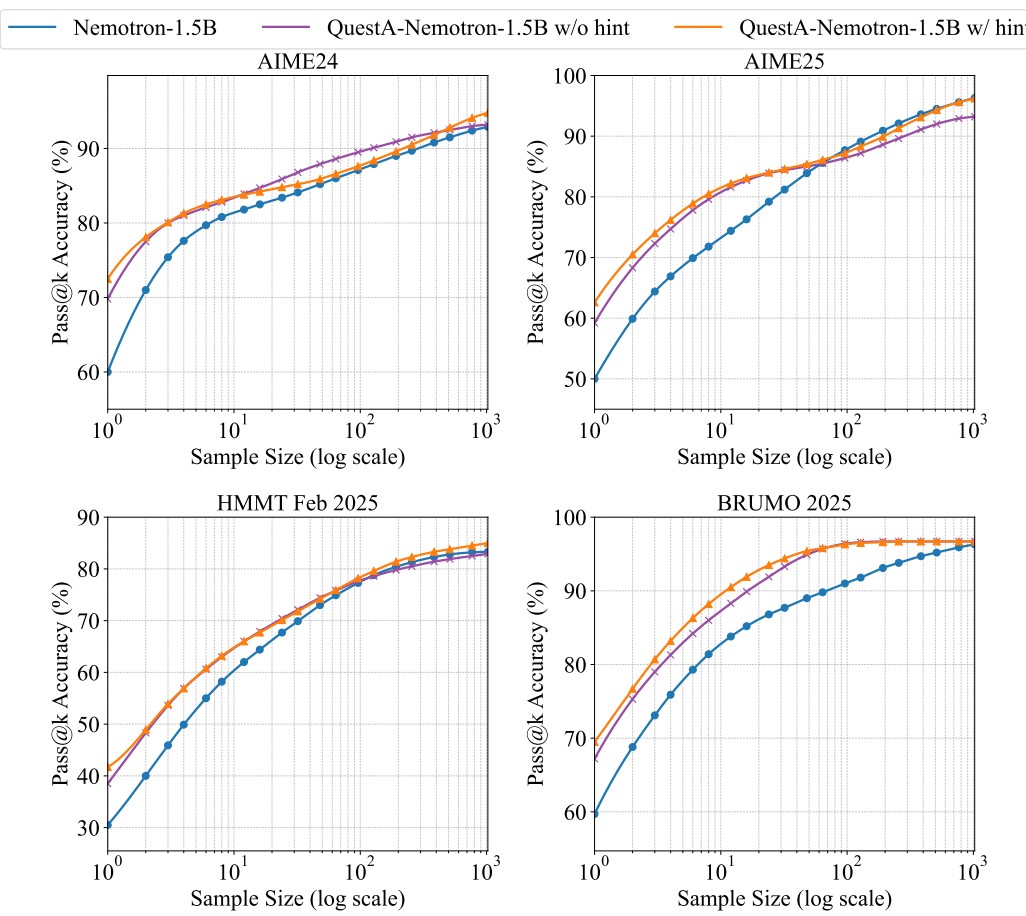

Figure 7: Pass@$k$ comparison on `Nemotron-1.5B` for RL *with* vs. *without* hints. Training with hints consistently dominates across $k$ and avoids the performance drop at larger $k$ seen in standard RL. Hints are used only during training; evaluation uses no hints.

We ablate the role of hints by training RL on `Nemotron-1.5B` with and without partial-solution hints. Here, QUESTA-Nemotron-1.5B *w/o hint* denotes RL on the same data and schedule but with the hint removed from the prompt; QUESTA-Nemotron-1.5B *w/ hint* uses identical settings except that the partial solution is provided as a hint during training. As summarized in Table 6, removing the hint still improves over the base model (average Pass@1: 53.14 → 60.44), but adding the hint yields a

further +2.82 average gain (60.44 → 63.26), with consistent improvements across all five benchmarks. The one using hint requires nearly half the number of steps compared to the one not using hint to achieve the same performance. Note that hints are used only during training; all evaluations are conducted *without* hints.

Figure 7 compares Pass@$k$ curves. The *w/ hint* model lifts the entire curve across $k$ and avoids the degradation at larger $k$ commonly observed in standard RL, while the *w/o hint* variant brings smaller gains that taper off as $k$ increases. A possible reason for this phenomenon is that, without hints, extremely difficult problems remain unlearned. Consequently, during reinforcement learning training, the model prioritizes improving performance on problems that have become relatively easier as training progresses. This leads the model to become overly confident, thereby reducing its Pass@k metric. In contrast, when hints are provided, the model still prioritizes learning more difficult problems—this is because such problems can provide effective learning signals.

## D.2   ABLATION STUDY WITH DIFFERENT MODELS

Table 7: Performance comparison on `DeepScaleR-1.5B`: Pass@1 (avg@32) across maths benchmarks. QUESTA consistently improves all tasks and raises the average by +6.50 points.

| Model | AIME24 | AIME25 | HMMT FEB 25 | Olympiad Bench | BRUMO25 | Avg |
|---|---|---|---|---|---|---|
| DeepScaleR-1.5B | 40.42 | 31.35 | 19.27 | 52.97 | 37.40 | 36.28 |
| QUESTA-DeepScaleR-1.5B | 49.16 | 35.94 | 21.77 | 58.69 | 48.33 | 42.78 |

Table 8: Performance comparison (Pass@1, averaged over 32 samples) showing the impact of QUESTA across benchmarks in other domains, including general knowledge, logic, and coding tasks. We observe minor cross-domain generalization on all these benchmarks, despite QUESTA being applied exclusively in the maths domain.

| Model | GPQA Diamond | Zebralogic | Code Contest All | Codeforces | LCB V5 202410-202502 | Avg |
|---|---|---|---|---|---|---|
| DeepScaleR-1.5B | 38.5 | 14.26 | 9.07 | 8.79 | 19.57 | 18.04 |
| QUESTA-DeepScaleR-1.5B | 39.2 | 14.98 | 10.1 | 8.9 | 20.9 | 18.82 |

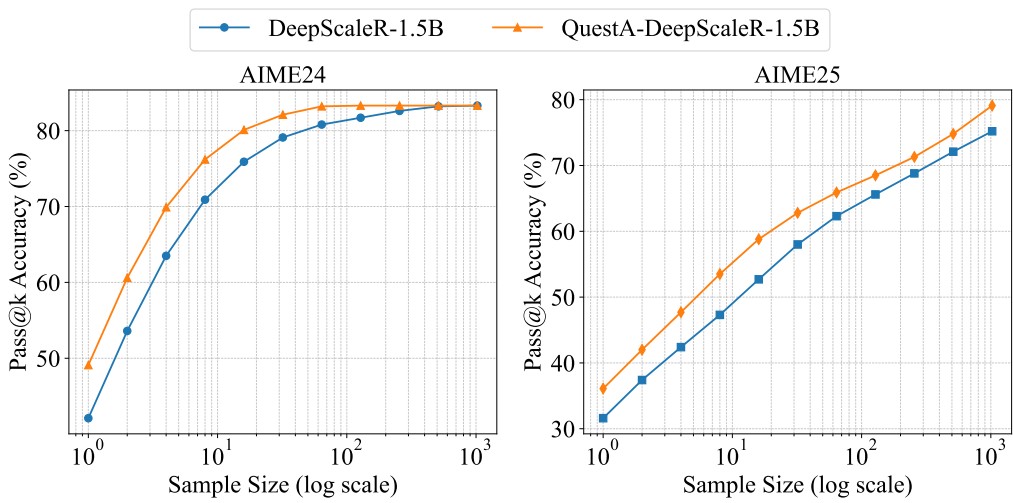

Figure 8: Pass@$k$ on `DeepScaleR-1.5B`: QUESTA raises the entire curve across $k$ and avoids the large-$k$ drop often seen in standard RL. The increasing gap with $k$ indicates improved sample diversity rather than overconfident collapse. No hints are used at evaluation.

We next examine model-family transfer by applying QUESTA to `DeepScaleR-1.5B`. We train for 750 steps on `DeepScaleR-1.5B Stage 2` Luo et al. (2025b) on the QUESTA first stage.

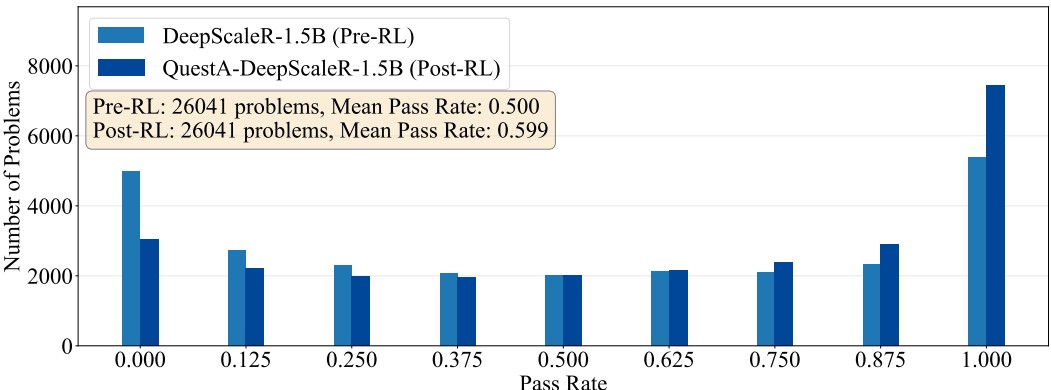

Figure 9: We conducted training of the DeepScaleR model employing the QUESTA with a dataset comprising 26,000 questions. The average pass rate was calculated from a sample of 8 instances. The initial graph represents the scenario without the incorporation of partial solutions, while the subsequent graph depicts the situation where partial solutions were included. The application of QUESTA significantly diminishes the incidence of unsolved or partially addressed problems within the training dataset. Concurrently, it has come to our attention that our previous method of data curation was not entirely accurate; in fact, the amount of data providing meaningful training signals is less abundant than anticipated, suggesting the potential for further refinement of the dataset.

As shown in Table 7, QUESTA-DeepScaleR-1.5B improves *every* maths benchmark over the base model, achieved an average improvement of 6%, indicating that the benefits of QUESTA are not tied to a single architecture.

Pass@k behavior mirrors these gains. In Figure 8, QUESTA-DeepScaleR lifts the entire Pass@$k$ curve across $k$ and avoids the degradation at larger $k$ reported in standard RL settings. The widening gap at larger $k$ suggests improved candidate diversity rather than overfitting to a single trajectory, consistent with our general pass@$k$ analysis in Appendix B.2. Complementing this, Figure 9 shows that on the 26K training set (evaluated *without* hints), mass shifts away from the 0/8–1/8 bins toward higher pass rates, reducing unsolved or partially solved cases. Hints are used only during training and are removed at evaluation time.

Beyond maths, Table 8 reports out-of-distribution (OOD) results on general knowledge, logic, and coding. QUESTA-DeepScaleR-1.5B achieves small but consistent gains (Avg: 18.04→18.82; +0.78), suggesting that the improved reasoning patterns transfer modestly beyond the training domain.

### D.3 SUPPLEMENTAL EXPERIMENT DETAIL

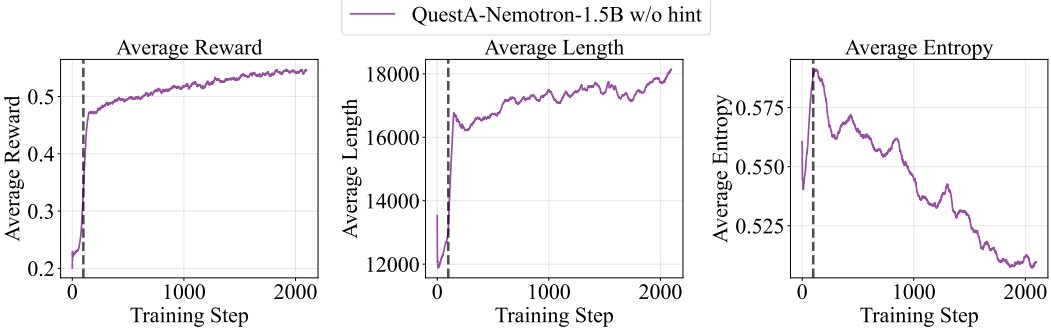

Figure 10: Training dynamics of QUESTA-Nemotron-1.5B w/o hint.

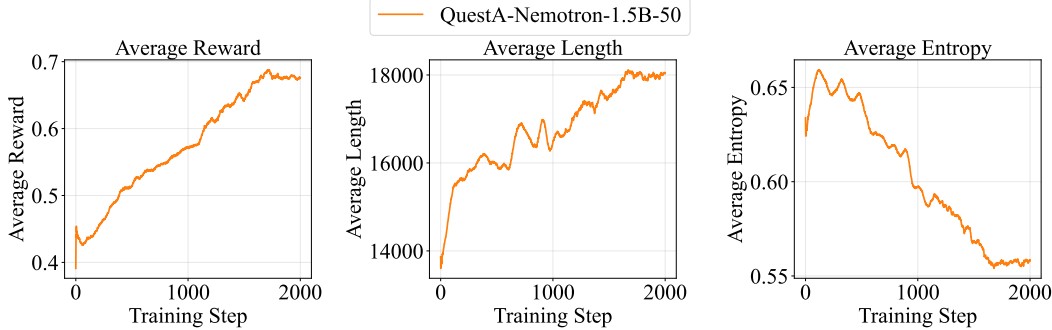

Figure 11: Training dynamics of QUESTA-Nemotron-1.5B-50.

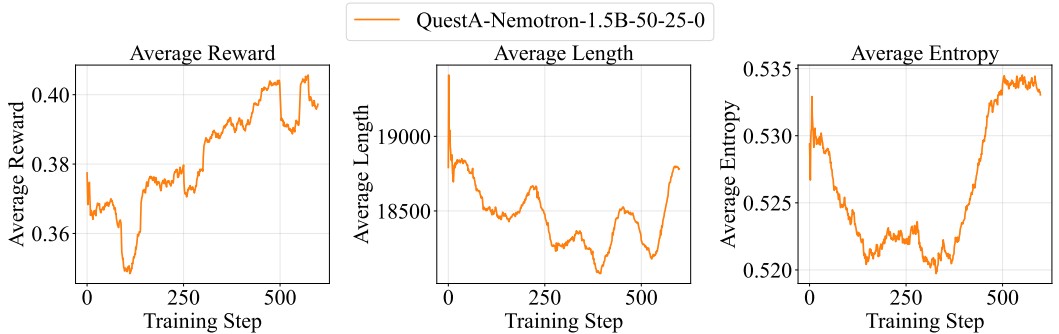

Figure 12: Training dynamics of QUESTA-Nemotron-1.5B-50-25-0.

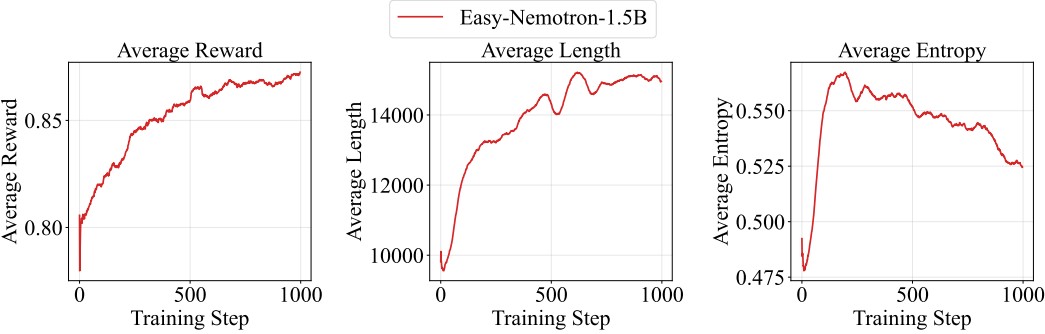

Figure 13: Training dynamics of Easy-Nemotron-1.5B.

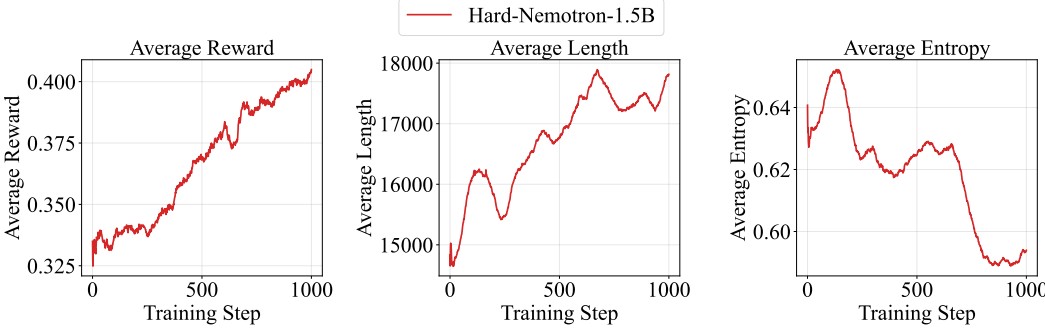

Figure 14: Training dynamics of Hard-Nemotron-1.5B.

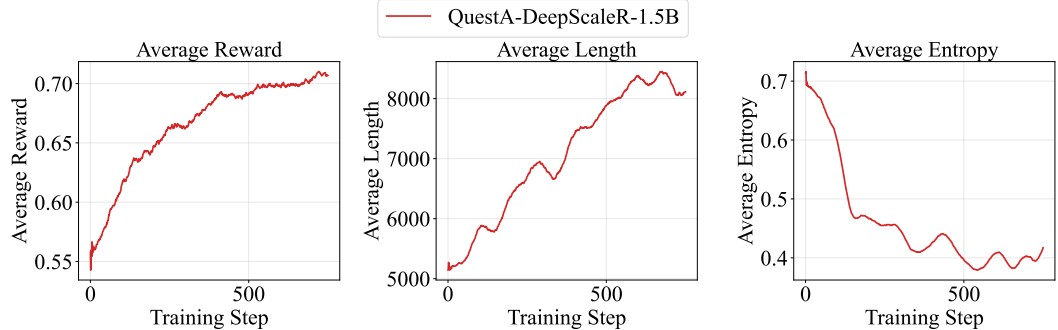

Figure 15: The training dynamics of QUESTA-DeepScaleR-1.5B. The first and second charts show the changes in the average reward and average inference length of the rollout samples that include all incorrect/correct ones, respectively. The third chart shows the average entropy excluding all incorrect/correct rollout samples.

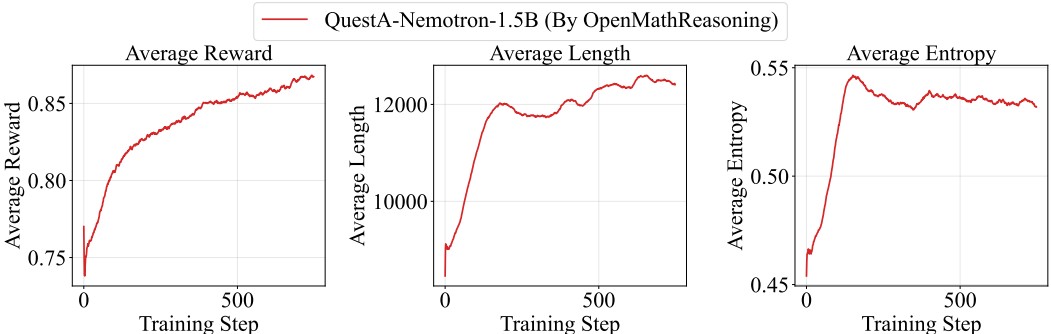

Figure 16: Training dynamics of QUESTA-Nemotron-1.5B on OpenMathReasoning (Moshkov et al., 2025). The first and second charts show the changes in the average reward and average inference length of the rollout samples that include all incorrect/correct ones, respectively. The third chart shows the average entropy excluding all incorrect/correct rollout samples. Dynamics closely mirror those on OpenR1-Math-220K, with no entropy collapse.

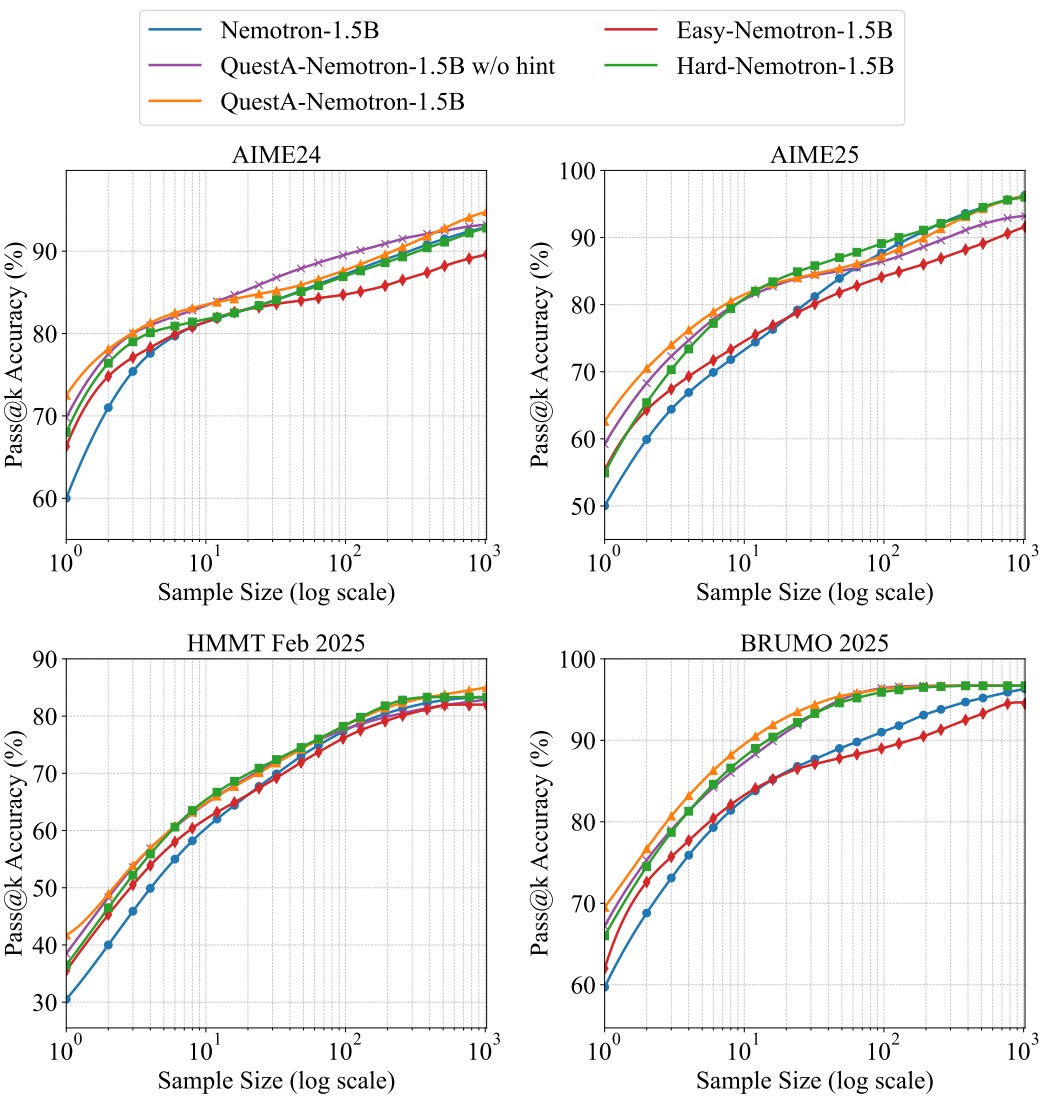

Figure 17: We compare pass@k curves of RLVR-trained models with and without QUESTA.

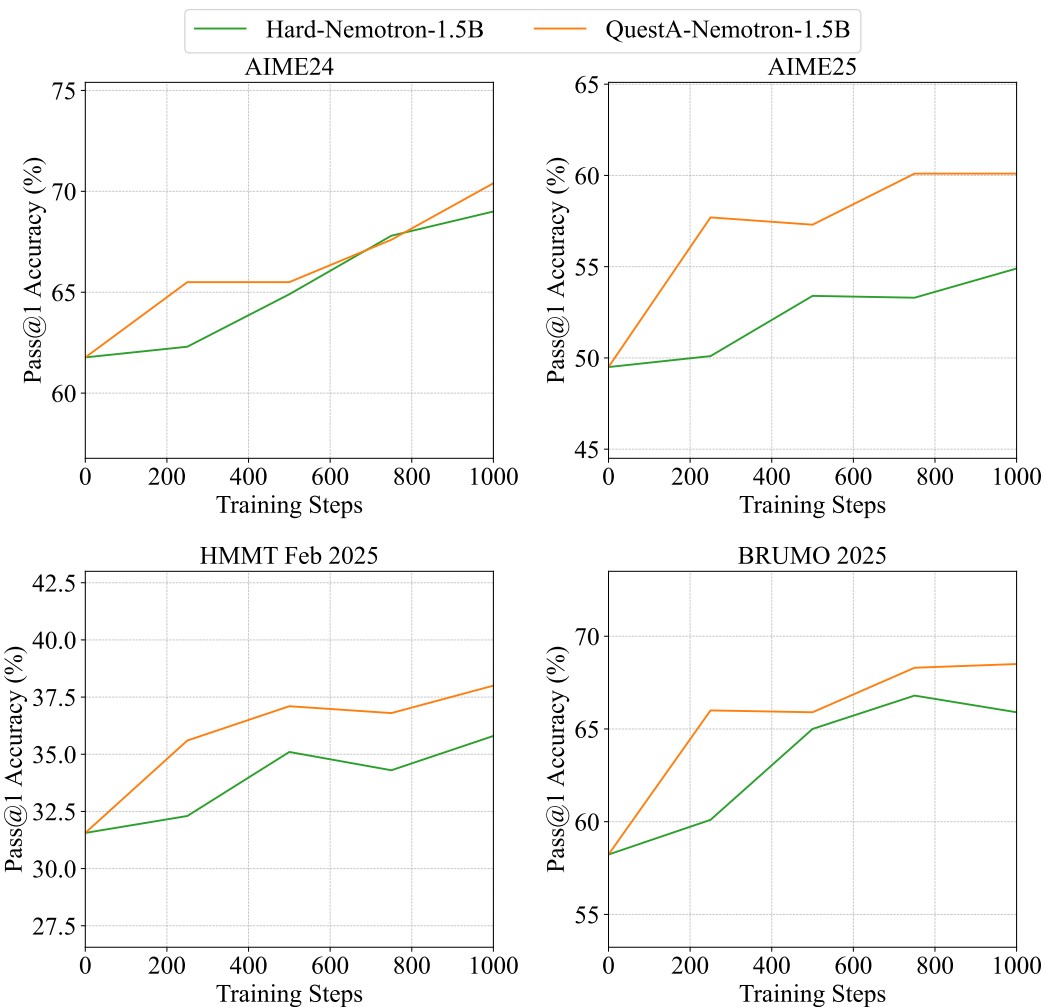

Figure 18: Comparison of RL training dynamics: Training with only hard problems (green) makes progress very slowly due to sparse rewards, while our method with partial solutions (orange) accelerates learning and consistently achieves higher accuracy across training steps.

