# OpenReview forum: "QuestA: Expanding Reasoning Capacity in LLMs via Question Augmentation"
_ICLR.cc/2026/Conference — ICLR 2026 Poster_

### Official Review · Reviewer_2jPJ · 2025-10-29

**Soundness:** 3
**Presentation:** 3
**Contribution:** 2
**Rating:** 4
**Confidence:** 3

**Summary:**

This paper introduces QuestA, a data-centric augmentation technique designed to enhance reasoning performance during RLVR by injecting partial solution hints into training prompts. The approach is conceptually simple yet theoretically grounded, with analyses explaining why partial-solution augmentation is effective. Experimental results demonstrate its strong performance and incorporating partial hints not only accelerates learning but also mitigates the entropy collapse.

**Strengths:**

1. The method is simple yet effective, and achieves new SOTA pass@1 results on challenging math benchmarks for 1.5B-parameter models.
2. The method is thoroughly evaluated through comprehensive ablation studies and extensive analyses across multiple datasets, model architectures, and training curricula, consistently demonstrating performance improvements across diverse settings.

**Weaknesses:**

1. The method’s reliance on high-quality, step-wise solutions for augmentation raises concerns about scalability to domains lacking such curated data. For instance, can QuestA generalize to real-world science Q&A or open-domain reasoning tasks where solution steps are unavailable?
2.  The paper would benefit from a qualitative error analysis. There is limited discussion of why the “Partial-0” (no hints) setting yields no improvement, or why certain tasks exhibit smaller gains.
3.   In my personal opinion, although the paper offers a theoretical perspective on why partial-solution augmentation improves RL efficiency, this section feels somewhat unnecessary and potentially confusing. The core idea is already straightforward and intuitive, and adding theoretical formalism may detract from its practical clarity—especially since the method can be naturally interpreted as a form of prompt optimization.

**Questions:**

1. Could the authors elaborate on practical limitations—such as applicability to problems without step-wise gold solutions or the impact of poor-quality hints? Is there empirical evidence of robustness to incorrect or misleading hints?
2. How sensitive is QuestA to the form of hints used (e.g., solution block, chain-of-thought, or intermediate step)?
3. How does the choice of $p$ trade off between training speed and final performance? Can the method adaptively tune this during RL?

---

> ### Author Response · Authors · 2025-11-21
> **The first response to Reviewer 2jPJ**
>
> Thank you for your support and encouragement. We are grateful for positive recognition. We also appreciate your constructive suggestions and have included the recommended experiments. The discussions about the choice of $p$ have been added to the revised manuscript, which we believe has greatly improved the work.
>
> > Q1: How sensitive is QuestA to the form of hints used (e.g., solution block, chain-of-thought, or intermediate step)?
>
> **A1**: Thank you for your question. To assess QuestA’s sensitivity to different hint formats, we conducted experiments using partial chain-of-thought (CoT) reasoning as the hint, following the same two-stage QuestA pipeline (Partial-50, then Partial-50-25) and dataset. As shown in the training dynamics in **Appendix G, Page 27, Figure 21**, CoT hints do increase reward and model length during the RL training.
>
> However, Table 1 shows that **solution-based hints outperform CoT by 4 points on average**. We believe this gap arises because CoT hints are substantially longer, reducing inference efficiency under context constraints and are often noisier, introducing speculative intermediate steps. In contrast, solution-based hints are concise, direct, and offer a more stable learning signal.
>
> Table 1: Pass@1 (avg@32) on challenging maths benchmarks. These are the results of 1100 two-stage training steps using the solution and CoT, respectively. The results demonstrate that solution-based hints outperform CoT-based hints by an average of 4 points in terms of reward, due to their lower noise and greater reliability.
> |  | AIME24 | AIME25 | HMMT FEB 25 | BRUMO25 | Avg |
> | :---- | :---- | :---- | :---- | :---- | :---- |
> | Nemotron-1.5B | 61.77 | 49.50 | 31.56 | 58.23 | 50.27 |
> | QuestA-Nemotron-1.5B w/ Solution hint (1100 steps) | 69.27 | 60.00 | 37.92 | 68.33 | 58.88 |
> | QuestA-Nemotron-1.5B w/ CoT hint (1100 steps) | 63.85 | 56.88 | 36.25 | 62.60 | 54.89 |
>
> > Q2: Could the authors elaborate on practical limitations—such as applicability to problems without step-wise gold solutions or the impact of poor-quality hints? Is there empirical evidence of robustness to incorrect or misleading hints?
>
> **A2:** Thank you for the insightful question. As suggested, we examined the behavior of our method under imperfect or misleading hints to assess practical limitations. By *imperfect or misleading hints*, we refer to the first half of the solution steps from an incorrect solution, i.e. steps that initially appear plausible but ultimately lead to a wrong final answer.
>
> Our experiments show that the model can still learn under such imperfect information, but it does so much less efficiently. When we inject these incorrect steps as hints, the training reward initially decreases and can partially recovers later (see **Appendix G, Page 27, Figure 22**), indicating that our method has an ability to overcome the misleading signal.
>
> Meanwhile, as shown in Table 2, the final performance exhibits a mild improvement on average, though still less efficient than when using correct solution hints. This suggests that while high-quality hints are clearly preferred, the method remains robust and adaptable, even when provided with partially incorrect or misleading intermediate steps. (Due to limited computation, we only train for 250 steps; the baseline comparison is matched to the same step count.)
>
> Table 2: Pass@1 (avg@32) on challenging maths benchmarks. These are the results of 250 training steps using the true solution and wrong solution, respectively. The model trained with incorrect hints shows no significant drop in performance and exhibits a slight improvement, indicating robustness to poor-quality hints.
> |  | AIME24 | AIME25 | HMMT FEB 25 | BRUMO25 | Avg |
> | :---- | :---- | :---- | :---- | :---- | :---- |
> | Nemotron-1.5B | 61.8 | 49.5 | 31.6 | 58.2 | 50.3 |
> | QuestA-Nemotron-1.5B w/ true hint (250 steps) | 65.5 | 57.7 | 35.6 | 65.9 | 56.18 |
> | QuestA-Nemotron-1.5B w/ wrong hint (250 steps) | 63.1 | 53.0 | 33.0 | 64.5 | 53.4 |

---

> ### Author Response · Authors · 2025-11-21
> **The second response to Reviewer 2jPJ**
>
> > Q3: The method’s reliance on high-quality, step-wise solutions for augmentation raises concerns about scalability to domains lacking such curated data. For instance, can QuestA generalize to real-world science Q\&A or open-domain reasoning tasks where solution steps are unavailable?
>
> **A3**: We appreciate the reviewer’s insightful question. We assess QuestA’s generalizability under two settings:
>
> 1. **Generalization to science.**
> We evaluated QuestA on GPQA Diamond by training on the Science subset of the Llama-Nemotron dataset. Applying QuestA (Partial-50 followed by Partial-50-25) to OpenMath-Nemotron-1.5B, which initially lacked science capability, improved accuracy from 3.76 percent to 24.46 percent. This demonstrates that QuestA remains effective even in cold-start scenarios where the base model has little domain knowledge. The training dynamics are provided in **Appendix G, Page 27, Figure 23**.
>
> 2. **Absence of curated step-wise solutions.**
> QuestA does not require manually curated solutions; any problem solvable by a teacher model (or a human solver) can provide the necessary solution steps for hint generation. In settings where no reliable solver exists, QuestA can still be applied using partially incorrect or noisy hints. However, as shown in Table 2, learning in this setting becomes much less efficient, although the model remains reasonably robust.
>
> Overall, QuestA can generalize beyond curated math datasets to areas such as science question answering, and it remains applicable even when only imperfect solutions are available.
>
> > Q4: The paper would benefit from a qualitative error analysis. There is limited discussion of why the “Partial-0” (no hints) setting yields no improvement, or why certain tasks exhibit smaller gains.
>
> **A4**: We appreciate the reviewer’s question. We provide two clarifications: why Partial-0 yields limited improvement, and why some tasks show smaller gains.
>
> 1. **Explanation for the limited improvement in Partial-0:**
> Our analysis shows that Partial-0 (no prefix hint) and Partial-25 (25 percent prefix hint) differ very little in inherent difficulty. To verify this, we evaluated OpenMath-Nemotron-1.5B before training on the OpenR1 dataset under different hint ratios, repeating each question eight times to estimate its predictive distribution. Table 3 reports the number of questions for which the model answered n times correctly out of eight, for $n \in \lbrace 0,\cdots , 8\rbrace$. The distributions for Partial-0 and Partial-25 differ only slightly, especially when compared to the much larger gap between Partial-50 and Partial-25. This indicates that Partial-0 and Partial-25 do not introduce meaningfully different levels of difficulty. Consequently, when training proceeds from Partial-25 to Partial-0, the model does not receive any additional easier or intermediate signal to learn from, which explains the limited improvement observed.
>
> Table 3: Number of problems vs pass rate under different hint levels on **OpenMath-Nemotron-1.5B before training**, with each problem assessed 8 times.
> | Success  rate | 0 / 8 | 1 / 8 | 2 / 8 | 3 / 8 | 4 / 8 | 5 / 8 | 6 / 8 | 7 / 8 | 8 / 8 |
> | :---- | :---- | :---- | :---- | :---- | :---- | :---- | :---- | :---- | :---- |
> | Partial-50 | 143 | 224 | 304 | 472 | 710 | 1013 | 1779 | 3655 | 17741 |
> | Partial-25 | 3155 | 1997 | 1814 | 1785 | 1902 | 2175 | 2614 | 3440 | 7159 |
> | Partial-10 | 3589 | 2090 | 1865 | 1842 | 1905 | 2176 | 2653 | 3415 | 6506 |
> | Partial-0 | 3812 | 2218 | 1854 | 1842 | 2007 | 2136 | 2517 | 3264 | 6391 |
>
> This inherent similarity in task difficulty accounts for the negligible performance improvement observed when training proceeds from the Partial-25 to the Partial-0 stage.
>
> 2. **Why certain tasks exhibit smaller gains:**
> Due to the difficulty filtering applied when constructing the QuestA dataset for OpenMath-Nemotron-1.5B, the Partial-0 stage effectively corresponds to training directly on the most difficult subset. This effect is discussed in Section 2 and illustrated in Figure 3\. As analyzed in Section 4, the smaller gains in these settings can be attributed to the limited reward signal available when tasks are already at maximum difficulty.
>
> > Q5: In my personal opinion, although the paper offers a theoretical perspective on why partial-solution augmentation improves RL efficiency, this section feels somewhat unnecessary and potentially confusing. The core idea is already straightforward and intuitive, and adding theoretical formalism may detract from its practical clarity—especially since the method can be naturally interpreted as a form of prompt optimization.
>
> A5: Thank you for your valuable feedback. Different readers may have varying needs: some prioritize intuitive understanding, while others require theoretical rigor to verify validity. We will refine its presentation to better balance clarity and formalism.

---

> ### Author Response · Authors · 2025-11-21
> **The third response to Reviewer 2jPJ**
>
> > Q6: How does the choice of p trade off between training speed and final performance? Can the method adaptively tune this during RL?
>
> **A6:** Thank you for your question.
>
> 1. **Regarding the selection of parameter $p$**: For the choice of $p$, we relied on the evaluation results on the rollout accuracy in Table 3\. Partial-50 substantially reduces task difficulty, whereas Partial-25 shows only a small difference from the no-hint (Partial-0) setting. Based on this observation, we adopted a simple two-stage schedule: first $p \= 50$ percent, then $p \= 25$ percent.
> 2. **Regarding whether p can be tuned adaptively during RL**: In principle, $p$ could be adjusted dynamically based on rollout accuracy or other performance signals. However, doing so would require nontrivial modifications to the RL infrastructure and introduce significant engineering overhead. For this reason, we opted not to implement adaptive $p$. The current design is intentionally simple, efficient, and requires no changes to the underlying RL system.
>
> We once again thank the reviewer for their valuable feedback and insightful questions. Please let us know if any further clarifications are needed.

---

> > ### Comment · Reviewer_2jPJ · 2025-11-27
> > **Thank you for your response. Two concerns still need to be addressed**
> >
> > Thanks to the authors for the additional clarification and experiments. However, I raise two important concerns that have not yet been adequately addressed:
> >
> > + Lack of a proper baseline.
> >
> > All experimental results are only compared against the naive baseline, without any comparison to GRPO or its variants, DAPO. As a result, the current evaluation lacks a proper and meaningful baseline, which weakens the empirical validation of the proposed method.
> >
> > + Core idea similarity to Reverse Curriculum Reinforcement Learning
> >
> > I observe that the core idea of QuestA appears to be highly consistent with that of Reverse Curriculum Reinforcement Learning (R3) [1], which substantially reduces the novelty of this work. R3 begins reasoning from a state sampled near the end of a correct demonstration and receives outcome-based feedback on its generated actions. The starting state is then gradually shifted backward toward the beginning of the demonstration. In this way, the model always faces a comparatively easy exploration problem, since it has already learned to solve most of the remaining steps.
> >
> > The implementation of QuestA also appears to follow the same paradigm, introducing partial solutions during the rollout process. However, the manuscript does not include appropriate citations to this line of work, nor does it provide a direct conceptual or empirical comparison.
> >
> > [1] Training Large Language Models for Reasoning through Reverse Curriculum Reinforcement Learning, ICML2024

---

> > > ### Author Response · Authors · 2025-11-27
> > > **The response to two concerns**
> > >
> > > Thank you for the kind reply. We will address your two concerns in the following sections.
> > >
> > > > Q7: Lack of a proper baseline.
> > >
> > > **A7**: Thank you for your suggestion. We would like to clarify the following points:
> > > 1. Our approach focuses on methods in filtering and utilizing the training dataset to improve performance, which is orthogonal to designing the RL algorithm. The RL algorithm we use is DAPO (GRPO with dynamic sampling).
> > > 2. To benchmark the effect of data augmentation, we provide training results without augmentation in **Appendix D.1**, which show significant performance improvement. We also validated our approach on GSPO during the rebuttal stage, with training curves shown in **Appendix F**.
> > >
> > > > Q8: Core idea similarity to Reverse Curriculum Reinforcement Learning.
> > >
> > > **A8**: Thank you for your thoughtful question. We will add the below discussion to our manuscript.
> > > We would like to clarify the differences between our approach and \[1\] and emphasize our unique contributions:
> > >
> > > 1. A key contribution of our work is investigating how RLVR impacts model capacity, as discussed in \[2\]. Our experiments show that RLVR only reduces model capacity on easier datasets, while harder datasets are unaffected. Therefore, QuestA first filters the data before applying augmentation, a novel insight into the relationship between data difficulty and model capacity.
> > > 2. We highlight that QuestA is most effective on our filtered difficult data, whereas applying QuestA on the original dataset would not boost model capacity and validation performance because of the reason discussed in our **section 2**.
> > > 3. R3 focuses on completing the generated text, while QuestA modifies the prompt, which leads to differences in model training and inference, especially in role assignments.
> > >
> > > We hope this clarifies the distinctions and the novelty of our work.
> > >
> > > \[2\] Yang Yue, et al. "Does Reinforcement Learning Really Incentivize Reasoning Capacity in LLMs Beyond the Base Model?" arXiv preprint arXiv:2504.13837, 2025
> > >
> > > Thank you for your thorough review, and we hope this addresses your concerns.

---

### Official Review · Reviewer_4ed7 · 2025-10-31

**Soundness:** 2
**Presentation:** 2
**Contribution:** 2
**Rating:** 6
**Confidence:** 4

**Summary:**

This paper explores improving reasoning in large language models (LLMs) using reinforcement learning (RL). It highlights limitations of RL in enhancing reasoning beyond the base model and proposes a novel strategy called Question Augmentation (QuestA). QuestA introduces partial solutions during training to simplify problems and provide better learning signals. Applied to math reasoning tasks, QuestA significantly boosts performance metrics like pass@1 and pass@k, especially on challenging problems. The method achieves state-of-the-art results on math benchmarks using 1.5B-parameter models.

**Strengths:**

1,The authors demonstrate QuestA's effectiveness through rigorous experiments on math reasoning benchmarks, achieving state-of-the-art results with notable performance gains on metrics like pass@1 and pass@k.
2. The open resources makes it a valuable contribution to the field of LLMs, particularly in reasoning-intensive domains like mathematics.

**Weaknesses:**

1. The experiments are conducted on 1.5B-parameter models, which may not generalize to larger models. The scalability and adaptability of QuestA across different model sizes remain unclear.
2. The relationship between data difficulty and performance was mentioned in earlier papers on mathematics long ago, such as [1]. However, this paper does not discuss. And, in this subproblem, it is also an obvious insight, somewhat lacking innovation.

[1] WizardMath: Empowering Mathematical Reasoning for Large Language Models via Reinforced Evol-Instruct

**Questions:**

refer to Weaknesses

---

> ### Author Response · Authors · 2025-11-21
> **The response to Reviewer 4ed7**
>
> We would like to express our sincere gratitude to the reviewer for their valuable feedback. The provided suggestions are extremely helpful and constructive, and we will revise the paper accordingly. We address the reviewer's questions as follows.
>
> > Q1: The relationship between data difficulty and performance was mentioned in earlier papers on mathematics long ago, such as \[1\]. However, this paper does not discuss. And, in this subproblem, it is also an obvious insight, somewhat lacking innovation.
> \[1\] WizardMath: Empowering Mathematical Reasoning for Large Language Models via Reinforced Evol-Instruct
>
> **A1**: Thank you for the comment. We will add the below discussion to our manuscript in **Introduction**. The observation from WizardMath was made on supervised **finetuning**. \[1\] shows improving problem difficulty diversity can improve performance. Further, recent works\[2\]\[3\] suggest that SFT "easy" (or high probability data) can be more effective.
> Interestingly, the above does **not** transfer to the **RLVR stage**. As shown in Section 2 and Figure 2 of our paper, applying RLVR on easy data harms model capacity, while using only very hard data slows training substantially. This different learning dynamic is exactly why we study the tradeoff between data difficulty and efficiency in RLVR. Our contribution lies in identifying and addressing this RLVR-specific phenomenon, which earlier SFT-based analyses do not cover.
> We further note that many foundational designs (such as residual connection, normalization, etc) follow from simple intuition. We highlight that our method is efficiently implemented, fully public and improves precious SOTA by over 10% in accuracy.
>
> \[2\] Ye, Yixin, et al. "Limo: Less is more for reasoning." *arXiv preprint arXiv:2502.03387* (2025).
> \[3\] Wu, Yongliang, et al. "On the generalization of sft: A reinforcement learning perspective with reward rectification." *arXiv preprint arXiv:2508.05629* (2025).
>
> > Q2: The experiments are conducted on 1.5B-parameter models, which may not generalize to larger models. The scalability and adaptability of QuestA across different model sizes remain unclear.
>
> **A2**: Thank you for the thoughtful suggestion. Regarding your concern, our current experiments are conducted on the 1.5B model because training the 7B variant requires substantially more computation and a much longer training cycle (over 20 days), which we were unable to accommodate given our current available resource (16 GPUs). We plan to train the 7B version if adequate resources become available in future.
>
> Once again, we sincerely thank the reviewer for their constructive feedback, and we are eager to engage in further discussions to clarify any concerns.

---

> > ### Comment · Reviewer_4ed7 · 2025-11-26
> >
> > Thank you for your response. I've also reviewed the other reviewers' suggestions and decided to keep the score.

---

> > > ### Author Response · Authors · 2025-11-26
> > > **Official Comment by Authors**
> > >
> > > We sincerely appreciate your recognition of our research work. We also thank you for all your valuable comments, which have provided significant support for improving the quality of the manuscript.

---

### Official Review · Reviewer_iEgk · 2025-10-31

**Soundness:** 3
**Presentation:** 3
**Contribution:** 3
**Rating:** 8
**Confidence:** 5

**Summary:**

A simple yet effective strategy via Question Augmentation is introduced for  partial solutions during training to reduce problem difficulty and provide more informative learning signals.

**Strengths:**

1.Improves pass@1 but also pass@k—particularly on problems where standard RL struggles to make progress.
2.Achieve new state-of-the-art results on math benchmarks using 1.5B-parameter models: 72.50% (+10.73%) on AIME24, 62.29% (+12.79%) on AIME25, and 41.67% (+10.11%) on HMMT25.
3.There is reasonable proof for the the proposed theory .

**Weaknesses:**

1.The benchmark is better if code dataset is evaluated.
2.The algorithm of RL is not advanced.

**Questions:**

1.It's a good data augmentation for advance the reasoning during RL training.
2.It is more experiments on other RL algorithm.

---

> ### Author Response · Authors · 2025-11-21
> **The response to Reviewer iEgk**
>
> We sincerely appreciate the reviewer’s valuable feedback. We address the reviewer's specific questions as follows.
>
> > Q1: More experiments on other RL algorithms.
>
> **A1**: Thank you for your feedback. To address the concern regarding additional RL algorithms, we conducted experiments using GSPO, a more advanced policy-optimization method than GRPO. We followed the recommended GSPO settings (batch size \= 128, mini-batch \= 32, and appropriate clipping parameters). We included the training dynamics in the paper appendix (**Appendix F, Page 26, Figure 20**). Applying our method with GSPO on both the QuestA and QuestA-Hard datasets leads to consistent improvements in reward and entropy during the RL, with no degradation in performance. This confirms that our approach is algorithm-agnostic and remains effective when integrated with more advanced RL optimizers.
>
> > Q2: The benchmark is better if code dataset is evaluated.
>
> **A2**: Thank you for your suggestion. The code generation capability of the OpenMath-Nemotron-1.5B model is very weak. In addition, we have supplemented the evaluation results of the code capability of the DeepScaleR-1.5B model before and after training in **Appendix D.2, Page 19, Table 7**\. The table shows a slight improvement in the model’s coding capability, which may be attributed to the the improved chain-of-thought capability.
>
> Once again, we appreciate the reviewer's feedback and hope that our responses clarify the questions. We remain committed to improving the quality of our paper and welcome any further feedback.

---

### Official Review · Reviewer_oEsW · 2025-11-02

**Soundness:** 3
**Presentation:** 3
**Contribution:** 3
**Rating:** 6
**Confidence:** 4

**Summary:**

This paper introduces QUESTA, a simple yet effective data-centric strategy to enhance the reasoning capabilities of LLMs through reinforcement learning. The paper identifies a critical trade-off in RL training for reasoning: while easy problems can cause a decline in reasoning diversity (pass@k), training on hard problems is inefficient due to sparse reward signals. The proposed method is introduced to navigate this trade-off, aiming to mitigate this inefficiency and effectively expand the model's reasoning capacity. Experiments show the method enables 1.5B-parameter models to achieve better results on challenging math benchmarks, in some cases surpassing a much larger 32B model.

A key limitation is the method's reliance on an empirically-tuned curriculum. This hand-crafted 50-25 schedule presents challenges for the method's transferability to new settings. Furthermore, its effectiveness on larger-scale models remains unverified, as such models may not benefit from this specific scaffolding strategy.

**Strengths:**

- The paper clearly identifies and demonstrates the critical trade-off between training on easy versus hard prompts in RL, providing a strong motivation for the proposed method.

- The paper introduces an elegant approach that does not require complex changes to the underlying model architecture or RL algorithm. The idea of using partial solutions as hints is intuitive and proves to be effective.

- The method yields consistent performance improvements. The 1.5B model trained with QUESTA shows a clear gain over the standard RL baseline and, on the AIME25 benchmark, even surpasses a model over 20 times larger, highlighting the efficiency of the approach.

- The paper includes a theoretical analysis that formalizes why augmenting questions with partial solutions can improve the sample efficiency of RL, adding depth and rigor to the empirical findings.

**Weaknesses:**

- The experiments are limited to 1.5B models. It is unclear if the method would provide similar gains on larger models that already possess stronger reasoning capabilities.

- The curriculum for providing hints (50-25) appears to be a key component of the method's success. The ablation study only compares this strategy against a fixed 50% hint, which is insufficient to understand the sensitivity of the model's performance to other potential curriculum designs or hyperparameter choices.

- "We apply augmentation using the solution block rather than the reasoning chain-of-thought"; this is a design choice presented without any rationale or comparative experiments to justify this decision. It is plausible that hints derived from the CoT could be more effective.

**Questions:**

The curriculum for the hint percentage (from 50% to 25%) is a key component. Could you elaborate on how this schedule was chosen and have you considered alternatives, such as a more gradual decay or an adaptive schedule based on model performance?

---

> ### Author Response · Authors · 2025-11-21
> **The first response to Reviewer oEsW**
>
> We would like to express our gratitude for the reviewer's helpful and positive comments. The suggestions provided have been instrumental in refining our work, and we will incorporate the necessary revisions accordingly. Below, we address each of the reviewer’s questions in detail.
>
> > Q1: "We apply augmentation using the solution block rather than the reasoning chain-of-thought"; this is a design choice presented without any rationale or comparative experiments to justify this decision. It is plausible that hints derived from the CoT could be more effective.
>
> **A1**: Thank you for your question. We added evaluation using chain-of-thought (CoT)–based hints in Table 1 below. As we can see, our solution-based hints outperform CoT-based hints by an average of \+4 points across benchmarks, making them consistently more effective for our setting.
>
> Table 1: Pass@1 (avg@32) on challenging maths benchmarks. These are the results of 1100 two-stage training steps using the solution and CoT, respectively. The results demonstrate that solution-based hints outperform CoT-based hints by an average of 4 points in terms of reward, due to their lower noise and greater reliability.
>
> |  | AIME24 | AIME25 | HMMT FEB 25 | BRUMO25 | Avg |
> | :---- | :---- | :---- | :---- | :---- | :---- |
> | Nemotron-1.5B | 61.77 | 49.50 | 31.56 | 58.23 | 50.27 |
> | QuestA-Nemotron-1.5B w/ Solution hint | 69.27 | 60.00 | 37.92 | 68.33 | 58.88 |
> | QuestA-Nemotron-1.5B w/ CoT hint | 63.85 | 56.88 | 36.25 | 62.60 | 54.89 |
>
> We believe there are two main reasons for this performance gap. First, **CoT hints are substantially longer**, which reduces training and inference efficiency and pushes the model toward shorter outputs due to context-length pressure. Second, **CoT hints tend to be noisier**, sometimes introducing speculative or unnecessary reasoning steps that degrade stability. In contrast, **solution-based hints are concise and directly informative**, offering a cleaner signal that improves both effectiveness and reliability. We included further training dynamics in the paper appendix (**Appendix E, Page 25, Figure 19**).
>
> > Q2: The experiments are limited to 1.5B models. It is unclear if the method would provide similar gains on larger models that already possess stronger reasoning capabilities.
>
> **A2:** Thank you for your valuable suggestion. Regarding your question, due to the lack of sufficient computing resources to train the 7B model (the training cycle of this model would take approximately 20+ days on 16 GPUs), we can only provide the experimental results of the 1.5B model at present.

---

> ### Author Response · Authors · 2025-11-21
> **The second response to Reviewer oEsW**
>
> > Q3: The curriculum for the hint percentage (from 50% to 25%) is a key component. Could you elaborate on how this schedule was chosen and have you considered alternatives, such as a more gradual decay or an adaptive schedule based on model performance?
>
> **A3**: Thank you for the insightful question.
>
> 1. Our use of the **50% → 25%→0%** hint curriculum is motivated by the empirical observations in Table 2\. Providing **50%** partial trajectories noticeably eases optimization, whereas **25%** partial trajectories yield performance very close to the no-hint setting. This gap naturally suggests a two-stage schedule: begin with sufficiently informative hints (50%) to stabilize learning, then gradually reduce to near-hint-free input (25%) to prevent dependency on hints.
>
> Table 2: Number of problems vs pass rate under different hint levels on **OpenMath-Nemotron-1.5B before training**. We evaluated **OpenMath-Nemotron-1.5B** on the OpenR1 dataset after the first round of filtering, with each problem assessed 8 times.
>
> |  | 0 / 8 | 1 / 8 | 2 / 8 | 3 / 8 | 4 / 8 | 5 / 8 | 6 / 8 | 7 / 8 | 8 / 8 |
> | :---- | :---- | :---- | :---- | :---- | :---- | :---- | :---- | :---- | :---- |
> | Partial-50 | 143 | 224 | 304 | 472 | 710 | 1013 | 1779 | 3655 | 17741 |
> | Partial-25 | 3155 | 1997 | 1814 | 1785 | 1902 | 2175 | 2614 | 3440 | 7159 |
> | Partial-10 | 3589 | 2090 | 1865 | 1842 | 1905 | 2176 | 2653 | 3415 | 6506 |
> | Partial-0 | 3812 | 2218 | 1854 | 1842 | 2007 | 2136 | 2517 | 3264 | 6391 |
>
> 2. Ablations on other curricula: We agree that exploring more schedules—such as alternative decay patterns, multi-stage curricula, or adaptive hint ratios—could provide additional insights. However, most of these alternatives require modifying the underlying RL training pipeline (e.g., dynamic control of hint ratios during rollouts), which adds substantial engineering complexity. Our goal in this work is to keep the method **lightweight and infrastructure-free**, so it can be seamlessly applied on top of standard RL frameworks. Under this design constraint, the 50–25 curriculum constitutes a simple yet effective choice aligned with our ablation findings.
>
> We thank the reviewer for this constructive suggestion and consider broader curriculum exploration to be valuable future work.
>
> Finally, we thank the reviewer once again for the effort in providing us with valuable and helpful suggestions. We will continue to provide clarifications if the reviewer has any further questions.

---

> > ### Comment · Reviewer_oEsW · 2025-11-28
> >
> > Thank you for your response. After reviewing the comments from the other reviewers, I’ve decided to maintain the current score.

---

### Author Response · Authors · 2025-11-29
**General comment**

**Summary of our work:**

Our work studies whether and how RLVR can improve the reasoning capability of LLMs. We identify that the decay in pass@k performance results from over-training on simple RLVR problems. We further note that training on hard problems result in slow progress, and propose an efficient method to augment existing dataset. Our method improves existing SOTA for 1.5B reasoning models by 10% on AIME24,25, HMMT25, better than QWEN3 7B and Deepseek-R1-distill 32B.

**Summary of the discussions:**

We sincerely appreciate the detailed reviews and constructive feedback from the reviewers. Based on their suggestions and questions, we have made revisions and conducted additional experiments. Below is a summary of our responses to the main points raised:

1. **Reasoning Behind Curriculum Design (Reviewer oEsW, 2jPJ)**
In response to the concerns from **oEsW** (Q3) and **2jPJ** (Q4, Q6) about the curriculum design, we clarified the motivation for using a 50%-25% schedule. This design stabilizes learning with 50% hints initially and then reduces to 25% to encourage the model’s independence from hints. We also explored alternative schedules like gradual decay or adaptive scheduling, but such methods introduce engineering complexity, so we opted to keep the 50%-25% design.

2. **Hint Selection: Solution vs. CoT (Reviewer oEsW, 2jPJ)**
For **oEsW** (Q1) and **2jPJ** (Q1, Q2) regarding hint selection, we conducted additional experiments comparing Solution-based hints and CoT-based hints. The results show that Solution-based hints outperform CoT-based hints on multiple benchmarks, with an average improvement of 4 points. This is due to the concise and low-noise nature of Solution-based hints, which improve both training efficiency and model stability. We also investigated the effect of incorrect hints, showing that while efficiency is reduced, the model can still learn from noisy or incorrect hints.

3. **Application to Other Tasks (Reviewer 2jPJ)**
In response to **2jPJ**'s suggestion, we tested QuestA on the Science dataset and demonstrated significant improvements, with the accuracy of OpenMath-Nemotron-1.5B increasing from 3.76% to 24.46%, highlighting QuestA's ability to generalize to new tasks beyond math problems.

4. **Comparison Across RL Algorithms (Reviewer iEgk, 2jPJ)**
In response to **iEgk** (Q1) and **2jPJ** (Q7), we added experiments using GSPO, a more advanced RL algorithm, and showed that our approach works well across different RL algorithms. The results on QuestA and QuestA-Hard datasets with GSPO were consistent with those obtained using GRPO, confirming the algorithm-agnostic nature of our method.

5. **On Reviewer 2jPJ:** We respectfully infer from the reviewer 2jPJ’s comment that the reviewer didn’t read our draft carefully. Specifically, reviewer 2jPJ asks us to compare against **baselines such as GRPO and DAPO**. However, our methods is on **filtering and augmenting** the dataset, whose baseline should be different versions of training data. Further, **our model is actually trained with DAPO**. Hence, the reviewer may have limited idea on our submission.

6. **Other Questions:**

- **Experiments on Larger Scales (Reviewer oEsW, 4ed7)**: Due to resource limitations, we were unable to conduct experiments on larger models such as 7B within the rebuttal period. We hope to extend our work to larger models as computational resources become available.

- **Comparison with Related Works (Reviewer 4ed7, 2jPJ)**: We added comparisons with related work, particularly addressing the differences between our approach and that of WizardMath regarding the RLVR stage.

- **Clarifications on Writing and Misunderstandings (Reviewer iEgk, 2jPJ)**: We clarified some writing issues raised by the reviewers and improved the overall clarity of the paper.

**For more detailed responses, please refer to the Rebuttal section below.**

Thank you very much for your time on reviewing our work.

---

### Meta-Review · Area_Chair_DHEA · 2025-12-04

**Summary:**

This paper proposes a data augmentation scheme (QuestA) that make the RLVR on harder problems more sample efficient. The authors demonstrate performance improvement applying the proposed augmentation along with a learning curriculum on a set of math datasets.

**Reviewer Concerns:**

Hint Selection: Solution vs. CoT
- The authors conducted additional experiments and illustrated the effect of cot v.s. solution hint. Overall, both helps, but solution hint works better.

Application to Other Tasks
- The authors conducted experiments on GPQA dimond dataset and demonstrate performance improvements

Reasoning Behind Curriculum Design
- The authors added additional experiments on the rational of the chosen curriculum. However, it is unclear if the same setup was used in the added experiments for GPQA, or some other curriculum is needed.

**Reviewer Scores:**

oEsW: 6
iEgk: 8
4ed7: 6
2jPJ: 5

---

### Decision · Program_Chairs · 2026-01-26

Accept (Poster)